# A lightweight and secure protocol for teleworking environment

**Fahad Algarni** [ID] [1]☯*, **Saeed Ullah Jan** [ID] [2]☯

1 Faculty of Computing and Information Sciences, University of Bisha, Bisha, Saudi Arabia, 2 Higher Education Department of Khyber Pakhtunkhwa, Govt College Wari (Dir Upper), Peshawar, Pakistan

☯ These authors contributed equally to this work.
* fahad.alqarni@ub.edu.sa

## Abstract

The Internet has advanced so quickly that we can now access any service at any time, from any location. As a result of this capability, People around the world can benefit from the popularity and convenience of teleworking systems. Teleworking systems, however, are vulnerable to a range of attacks; as an unauthorized user enters the open communication line and compromises the whole system, that, in turn, creates a big hurdle for the teleworkers. Professional groups have presented numerous mechanisms for the security of teleworking systems to stop any harm, but there are still a lot of security issues like insider, stolen verifier, masquerade, replay, traceability and impersonation threats. In this paper, we propose that one of the security issues with teleworking systems is the lack of a secure authentication mechanism. In order to provide a secure teleworking environment, we have proposed a lightweight and secure protocol to authenticate all the participants and make the requisite services available in an efficient manner. The security analysis of the presented protocol has been investigated formally using the random oracle model (ROM) and ProVerif simulation and informally through illustration/attack discussions. Meanwhile, the performance metrics have been measured by considering computation and communication overheads. Upon comparing the proposed protocol with prior works, it has been demonstrated that our protocol is superior to its competitors. It is suitable for implementation because it achieved a 73% improvement in computation and 34% in communication costs.

**Data Availability Statement:** All relevant data are within the manuscript and its Supporting Information files.

**Funding:** The authors are thankful to the Deanship of Scientific Research at University of Bisha for

## 1 Introduction

The facility provided to someone to accomplish their assigned responsibilities remotely through the Internet, e-mail, chat, video conferencing, or other platforms is called teleworking. The convenience of working in a remote work environment through online meetings, chat, video conferencing, instant messaging, multimedia document collaboration, and coordination among workers worldwide has drawn the attention of researchers into the field of telework [1]. And for the last three to four years, particularly during the COVID-19 pandemic, there has been a marked increase. The incredible and dispersed organizational controls associated with telework inevitably lead to an increase in information security threats. For instance,

supporting this work through the Fast-Track Research Support Program.

**Competing interests:** The authors have declared that no competing interests exist.

workers who opt to work from home are unable to guarantee that their living quarters meet the bare minimum security standards. Moreover, some companies need to create a telework security strategy that lays out the expectations, limitations, and duties of teleworkers in terms of preventing and handling security events. Organizations may be more vulnerable to network security threats in these circumstances [2]. Therefore, secure authentication and cross-verification of all participants are mandatory to ensure information security.

With the widespread developments in networking technology, unified communication, and the output produced by the Internet of Things (IoT), everyone may now do duties outside of the office with greater ease due to the Internet. To save time and money, competent and well-trained individuals may efficiently and flexibly provide services remotely from homes and other suitable locations. Their production will grow because of fewer workplace distractions, greater autonomy, and balanced work, saving businesses money and resources while requiring less real estate expenditure. However, this revolution in the business and technology sectors also creates constraints by forcing firms to develop and adapt, mimicking corporate trends and continuous improvement in their IT and communication systems. It aims to expand resources by developing the infrastructure to minimize human involvement [3], as show in the Fig 1.

Conversely, lessening the human element might enhance digital work administration; however, managing the technological safety of all modern traffic, whether inside or outside the company, is difficult as secure communication is one of the most important factors in ensuring

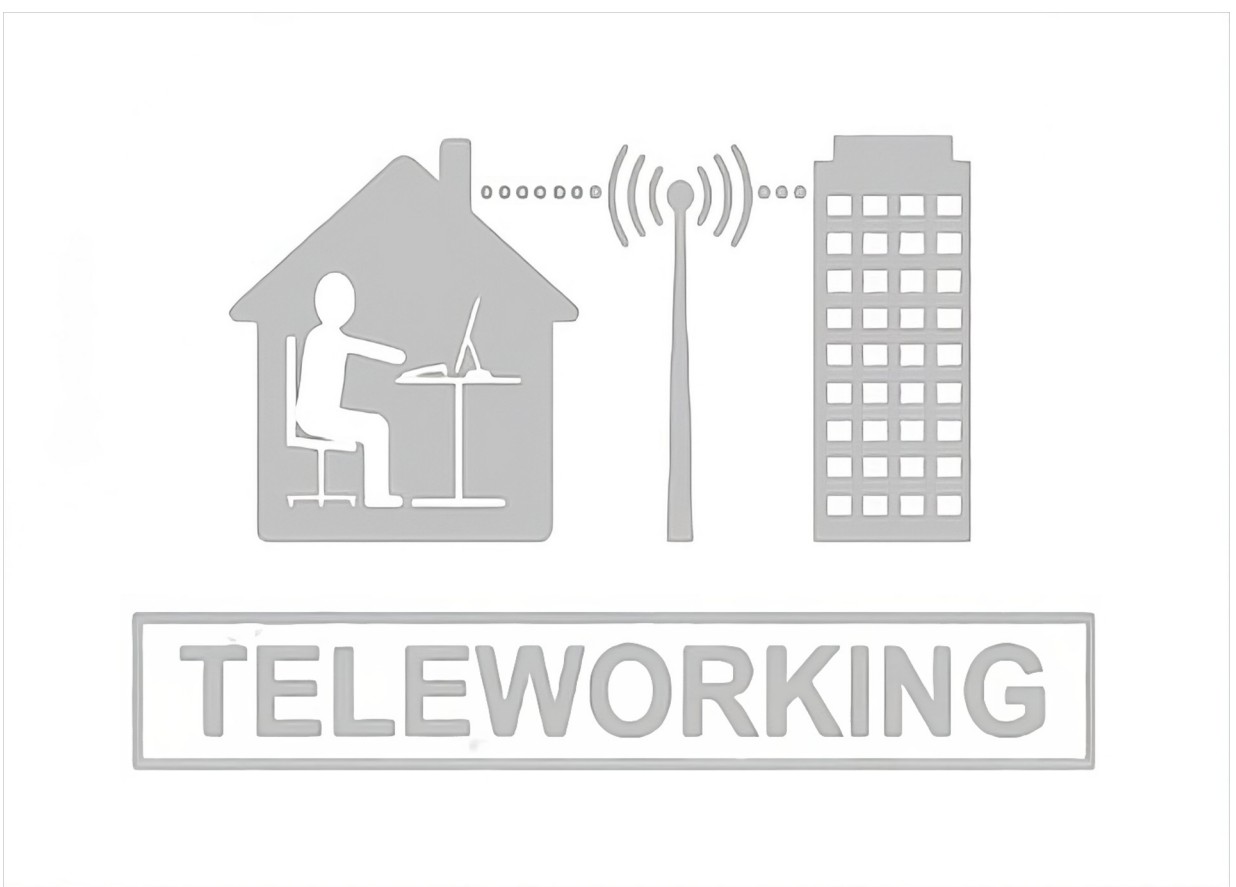

**Fig 1. A Sketch of teleworking environment.**

data protection, accessibility, secrecy, and authenticity. Challenges with teleworking include monetary harm and data vulnerabilities that expose business data [4].

Conventional security methods that relied on a dynamic framework did not function well; teleworkers still need a safe and secure workplace even with powerful intrusion detection systems (IDS), firewalls, sophisticated encryption, anti-virus software, and other safety precautions [5]. The increasing number of skilled workers and teleworking outside the business infrastructures using capabilities offered by internet service providers (ISP) make the security of transferred information even more important. Companies that let staff work on assignments and operations from home to increase efficiency have put their corporation's digital security at risk [6]. Businesses must be aware of the dangers of leaving themselves open to various attacks. To ensure the presence of risks for all parties concerned, they must fortify themselves with resources and proactive strategies [7].

## 1.1 Motivation

By creating and maintaining a telework security plan, protecting conversations and data saved on client devices, and assuring that remote servers/peripherals are appropriately accessed, a robust authentication of all participating entities is needed that can guarantee secure communication—also, keeping in view the fundamental security features like privacy protection, remotely working in a secure environment, time management, non-physical work environments at home, and intrinsic and extrinsic workload, etc. motives us to design a security system (remote authentication of all the participants in a secure manner) for such a vulnerable environment. And how various genders adapt to a work-life balance without annoying the traditional family culture. The proposed security system could help companies to reorganize their structures with greater flexibility.

## 1.2 Challenges and contributions

Employees who work remotely may be isolated from one another, which could inhibit their ability to react to security threats. Furthermore, there may be dangers associated with needing to have control over how sensitive data is used, stored, and deleted across different applications. Insufficient data security in teleworking environments can lead to data breaches, where hackers can take advantage of weak authentication schemes and compromise the confidentiality, integrity and authorization of data. Weak authentication can create severe repercussions from such breaches, including malware attacks and harm to one's reputation (traceability and unprotected privacy). To combat these security challenges, some governments and corporations have restricted the physical involvement of their workers while keeping their output the same. These organizations have been responsible for providing a secure environment to their workers for working from home in the teleworking environment, which will rely heavily on the Internet. So far, this research offers a security mechanism for a teleworking environment that tackles the aforementioned major issues and challenges of security. In this regard, we have designed a strong authentication for the remote monitoring of teleworkers, which offers protection against unprotected connections. The main contributions of this research work are as follows:

- To propose a lightweight and secure authentication protocol that can protect critical resources of the teleworking environment and mitigate all known threats due to giving unsecured external access to critical data/resources.

- To design a protocol with lightweight operations causing no delay in responding to security vulnerability and offering low computation and communication costs and robust security.

- To verify the security of the proposed authentication protocol both formally and informally by showing a delicate balance of security with performance, as these are opposing features often missing in previous protocols.

- To comparatively analyze it with state-of-the-art works in terms of security functionalities, performance metrics, communication, and computation overheads.

By successfully conducting the proposed research work, the following questions that a layperson can raise will be answered. Recently, people haven't felt secure due to the availability of strong adversaries–they didn't work remotely in a safe environment.

- How can a teleworking environment be secure?

- How can the attention of skilled people be catered to?

- How can it take less time for greater output?

- How fast can the work be done?

- How can energy-saving techniques for an organization be materialized?

The remaining paper is organized as in section 2, which contains the literature survey, we have also presented reviewe analysis of baseline scheme and cryptanalyzed it. The result of cryptanalysis shows that the scheme suffers from insider, bias, inaccuracy, and heavyweight; section 3 confesses the system model, threat models, design goals and key highlights; section 4 demonstrates the proposed authentication scheme, and in 5, we analyzed the security of the proposed protocol both formally using the random oracle model (ROM) and ProVerif simulation and informally using attacks' discussion; section 6 contains performance measurement of the proposed scheme, and in section 7, we have concluded the work.

## 2 Literature survey

Gupta et al. [8] identified drawbacks in an identity-based protocol used for remote working. They identified impersonation and insider attacks as the major loopholes. After that, [8] presented an improved ECC-based authentication scheme for mobile devices. However, [8] shows that the plan is lightweight and robust. Salami et al. [9] demonstrated that remote authentication could ensure any transaction's availability, non-repudiation, and integrity. And the online task is accomplished by many businesses through cloud-based computing. In this scenario, thousands of devices accessed cloud servers remotely through low-capacity mobile devices. Such devices are susceptible to potential threats and need rigorous attention. So far, they [9] have proposed a multi-level remote authentication protocol to prevent misusing exchanged information. Ahn et al. [10] argued that telework is a practical working platform that offers stakeholders a more efficient way of working.

It should be noted that the protection of exchanged information is much needed in this regard [11] have proposed near-field communication (NFC) based authentication protocol for teleworking. They claimed that the privacy of a remote user is a dire need of the day, and most of the work done by researchers has never tackled this major issue; the schemes available in the literature also failed to provide secure services to teleworkers. They [11] claimed that their security framework offers resistance to insider and impersonation threats and provides anonymity, untraceability, and forward secrecy.

With the rapid advancement in technology, which is becoming more mature, people's schedules are also getting more complicated. Right from dawn, their pursuit starts quite unaware of their immediate surroundings, including the household. They lock the door, hoping to control it from the workplace. This, however, is not immune to attacks from hackers

who may easily rob the valuables. To overcome these impending threats, we have attempted to connect all valuables in the household so they can be safely monitored/ controlled via mobile. Jan et al. [12,13] proposed a remote user's security mechanism for alleviating desynchronization attacks. They claimed that in most protocols, the random numbers generated at one end couldn't update their corresponding peers, creating a desynchronization flaw. To tackle such an issue, they efficiently mitigated it by saying that if one participant sent a random number to other participants, and the central server failed to verify its randomness, it means that someone had tempered the message and considered it a potential threat, promptly discarded and terminate the process. And if, for example, one participant sent a random number to some other participants, and the $\mathcal{A}$ captured it from the open line, the communicating parties, in this regard, don't believe in single running because all the participants must first agree upon a single key, then start communication. Their scheme fantastically highlighted the desynchronization issue; however, their scheme's computation cost is much higher due to modular exponentiation.

A three-factor symmetric key-based scheme has been presented by Zeeshan et al. [14] for telecare medicine information systems. The protocol suggested by [14] has reliably provided mutual authentication and perfect forward secrecy. However, it is weak against man-in-the-middle and session key disclosure attacks. Amin et al. [15] proposed a security framework for IoT in distributed cloud computing. Their scheme offered mutual authentication and could withstand impersonation attacks; however, they suffered from traceability attacks. Chaudhry et al. [16] demonstrated a protocol for distributed cloud computing. Their protocol has many merits, including its resisted Ephemeral Secret Leakage (ESL) and impersonation attacks. Also, their scheme competently provides perfect forward secrecy and mutual authentication. However, they forgot to mention the revocation/reissue phase, which is vital to security. Wu et al. [17] and Jia et al. [18] proposed a scheme for edge computing working for remote users. Their presented scheme's security and privacy-related security protocol securely provides mutual authentication and withstands password guessing and brute force attacks. However, it failed to resist man-in-the-middle attacks and didn't provide perfect forward secrecy.

Gope et al. [19] proposed a security protocol for the remote monitoring of an entity using a wireless sensor network. They argued that remote user authentication in a resource-limited environment is a critical task, and such paramount security concerns can only be handled by first efficiently authenticating the user and then starting data transmission. However, due to using symmetric cryptography and encryption/decryption functions, their lightweight claim needs to be made more explicit. Encryption/decryption is unsuitable for such a resource- and bandwidth-limited environment. At the movement, Shafiq et al. [20] designed an ECC-based lightweight authentication framework for authenticating a user remotely. But when an attacker steals the smart card, which is the primary entity in their scheme, they can quickly launch stolen-verifier and ESL attacks on their security mechanism. Taher et al. [21] proposed a three-factor authentication scheme for a remote user for IoT using WSN. They used AVISPA for simulation, BAN logic for hash code security checking, and fingerprint for additional security. However, the offline password-guessing attack has been noted in their scheme because when an $\mathcal{A}$ chooses an identity, they can quickly become successful for limited guesses.

Challa et al. [22] presented a framework for a heterogeneous-based cyber-physical system using IoT. They argued that cyber attacks were challenging when the number of IoT increases for the physical phenomenon, and such challenges couldn't be detected easily. Employing an efficient security system makes anonymity, privacy, and secure information broadcasting more straightforward to tackle. However, a signature-based scheme is not feasible for resource- and power-limited IoT. Similarly, if an attacker has stolen the smart card of a system, they can quickly figure out the internal credentials from it. Therefore, the proposed scheme

still needs to deliver secure services for the system. Wazid et al. [23] proposed an ECC-based authentication scheme for a smart home environment. Their scheme seriously tackled the issue of replay and clock synchronization attacks, which they have noticed in many security frameworks. However, in the second public channel transmission, the gateway node key ($SK_{GS}$) is transmitted openly, which the attacker can easily capture and identify many other credentials for DoS attack. Shuai et al. [24] proposed a robust remote monitoring authentication system using a symmetric key cryptographic method. The 1024-bit key is unsuitable for such a resource-limited and low-power IoT.

Oh et al. [25] presented a scheme for IoT-based smart homes. They used a lightweight asymmetric cryptographic method for designing their scheme. Their scheme has fantastically achieved its goals for the remote monitoring of intelligent gear installed in smart homes online through the internet. However, their scheme is unsafe against privileged insider and stolen verifier attacks because when an adversary steals the mobile device, the internally stored credentials can easily be identified and later used for malicious deeds. Ding et al. [26] designed a scheme using the fuzzy extractor method for securing user biometrics. However, their scheme is not safe against online/offline password-guessing attacks.

Kamble et al. [27] proposed a provable secure protocol for a tale-medicine information system using the chaotic map method. They have analyzed the security of their protocol through BAN logic and AVISPA simulation toolkit and claimed that their scheme has successfully preserved the privacy of users. However, using a chaotic encryption method, which is based on floating calculation, that in turn makes the hardware implementation difficult compared to AES and DES, which need integer operations. Meshram et al. [28] designed a scheme human-centred IoT system using the Quantum Chebyshev Chaotic (QCC) Maps method. They demonstrated that modelling analysis for IoT is much needed because of different human behaviour. Using encryption IoT can tackle the issue of human behaviour over IoT, but bilinear maps create hurdles while implementing these encryption-based security models. To mitigate this flaw, they have proposed the Quantum Chebyshev Chaotic (QCC) Maps method for the HC-assisted IoT system. However, due to using the Computational Diffie-Hellman [29] method for key exchange, their scheme is suitable for single-party authentication; when the number of participants increases, their scheme doesn't show efficiency. Another scheme [30] based on Fractional Chebyshev Chaotic Map-Based was also presented for the HC-IoT system. Upon going to check the protocol round-trips in the login authentication phase, it has been observed that in the first round trip, the identity is transmitted openly, which an attacker can catch and launch DoS and replay attacks later on. In [31], they used a digital signature technique for an HC-assisted IoT system. However, they didn't tell the reader about what type of verification their algorithm will perform, either one-to-one or bach.

## 2.1 Review analysis of baseline scheme

Recently, [17] proposed a novel authentication scheme based on the bilinear mapping technique. They have taken two groups, namely $|G|$, $|G_T|$, and $Z_p^*$, of key sizes 1024, 1024, and 160 bits, respectively. They have designed their scheme using SHA-256, symmetric encryption/decryption, and biometric Gen(.)/Rep(.) functions. Their scheme consisted of two phases, i.e., registration and login and authentication. The registration phase is accomplished at the following points:

1. The user selects identity $MID_i$, and transmits it to the registration center.

2. The RC chooses $r_i$, $x_i$, computes $TMID_i = h(MID_i||r_i)$, $B_i = TMID_i \oplus h(K_{RC}||x_i)$, stores $\{B_i, x_i\}$ and transmits $\{TMID_i, B_i, h(.)\}$ back towards the user smart card.

3. The user enters his/her $PW_i$, generates biometrics $BIO_i$, chooses $n_i$, computes $Gen(BIO_i) = (\sigma_i, \tau_i)$, $C_i = n_i \oplus h(MID_i||PW_i||\sigma_i)$, $Auth_i = h(MID_i||PW_i||\sigma_i||n)$, $TMID_i^* = TMID_i \oplus h(n||PW_i||\sigma_i)$, replace $TMID_i$ with $TMID_i^*$ and injects $\{C_i, Auth_i, Gen(.), Rep(.), \tau_i\}$ in the memory of mobile smart card.

4. Now, the server first chooses identity and transmits it to the registration center.

5. The RC also chooses two orbitaray numbers $r_j$, and $x_j$, computes $PSID_j = h(SID_j||r_j)$, secret key of server $K_S = h(h(SID_j||K_{RC})$, $Q_j = h(SID_j||x_j)$, $F_j = r_j \oplus Q_j$, stores $\{F_j, PSID_j, x_j\}$ and transmits $\{K_S, r_j\}$ back towards the server for also storing in its memory.

The login and authentication phase of the scheme [17] takes the following round-trips to complete. These are as follows:

1. The user insert their smart card, provides $MID_i$, $PW_i$, generates $BIO_i'$ and computes $Rep(BIO_i', \tau_i) = \sigma_i'$, $n_j' = C_i \oplus h(MID_i||PW_i||\sigma_i')$, checks $Auth_i = h(MID_i||PW_i||\sigma_i'||n_i')$, if confirmed, user choses a number $r_1$, time $T_1$, computes $TMID_i = TMID_i^* \oplus h(n_i||PW_i||\sigma_i')$, $r_1' = r_1 \oplus h(SID_j||TMID_i)$, $D_1 = SID_j \oplus h(TMID_i||T_1)$, $D_2 = h(SID_j||TMID_i||r_1||T_1)$ and transmits $\{r_1', B_i, D_1, D_2, T_1\}$ towards RC over a public network channel.

2. The RC checks the time space $T_1 - T_c \leq \Delta T$, computes $TMID_i = B \oplus h(K_{RC}||x_i)$, $SID_j = D_1 \oplus h(TMID_i||T_1)$, $r_1 = r_1' \oplus h(SID_j||TMID_i)$, $D_2' = h(SID_j||TMID_i||r_1||T_1)$, checks $D_2'? = D_2$, if becomes successful, RC selects a random number $r_2$ and time $T_2$, computes $D_3 = h(PSID_j||r_2) \oplus h(h(SID_j||K_{RC}))$, $Q_j = h(SID_j||x_j)$, $r_j = F_j \oplus Q_j$, $TMID_i' = h(PSID_j||r_j) \oplus TMID_i$, $D_4 = h(h(PSID_j||r_2||SID_j||T_2)$ and transmits $\{r_1', T_2, TMID_i', D_3, D_4\}$ towards server over a public network channel. 4

3. The server checks the time validity $T_2 - T_c \leq \Delta T$, computes $h(PSID_j||r_2) = D_3 \oplus h(K_S)$, $D_4' = h(h(PSID_j||r_2||SID_j||T_2)$, verify $D_4' = D_4$, if passes, selects a random number $r_3$ and time $T_3$ and computes $PSID_j = h(SID_j||r_j)$, $TMID_i = TMID_i' \oplus h(PSID_j||r_j)$, $r_1 = r_1' \oplus h(SID_j||TMID_i)$, $SK = h(h(PSID_j||r_2)||r_1||r_3||TMID_i)$, $r_3' = r_3 \oplus h(r_1||SID_j)$, $D_5 = h(r_1||r_3||T_3)$, $D_6 = h(SK||h(PSID_j||r_2||r_1)$ and transmits $\{r_3', T_3, D_5, D_6\}$ back towards RC over an open channel.

4. The RC verify the time threshold, selects $T_4$, computes $r_3 = r_3' \oplus h(r_1|\backslash SID_j)$, $D_5' = h(r_1||r_3||T_3)$, verify $D_5' = D_5$, if validated, computes $D_7 = h(PSID_j||r_2) \oplus h(r_1||TMID_1)$, $D_8 = h(h(PSID_j||r_2||r_3||T_4)$ and transmits $\{r_3', T_4, D_6, D_7, D_8\}$ to user over open channel.

5. The user check time space $T_4 - T_c \leq \Delta T$, computes $r_3 = r_3' \oplus h(r_1||SID_j)$, $h(PSID_j||r_2 = D_7 \oplus h(r_1||TMID_i)$, $D_8' = h(h(PSID_j||r_2||r_3||T_4)$, checks $D_8' = D_8$, if validated, computes $SK = h(h(PSID_j||r_2)||r_1||r_3||TMID_i)$, $D_6' = h(SK||h(PSID_j||r_2)||r_1)$, confirms $D_6' = D_6$, if matched, keep SK is secret session key for upcoming communication.

## 2.2 Cryptanalysis of baseline scheme

Upon thoroughly analyzing [17], the following vulnerabilities have been noticed:

**1) *Prone to Privileged Insider Threat*:** Many random numbers are extracted in each round trip of the protocol, which has a big chance for privileged insider threats. Similarly, in the scheme [17], the parameters stored in the smart card are $\{C_i, Auth_i, Gen(.), Rep(.), \tau_i\}$ in which a privileged user can select a random number $r_A$, computes $C_A = r_A \oplus MID_i$, $Auth_i = MID_i \oplus C_A$ and $TMID_A = MID_A||PW_A \oplus C_A$. After finding an identity, he/she can then easily launch a privileged insider attack. If we consider the same attack for the server, the credentials stored are $\{r_j, K_s\}$, whereas $r_j$ is a random number while $K_s = h(SID_j||s)$, which a privileged user can find easily. Therefore, [17] is prone to privileged insider threats.

**2) *Bias*:** The protocol presented in [17] minimizes biometric demographic bias for such a resource-limited environment. The biometrics used in [17] demonstrate notable variations in their functionality while engaging with distinct user demographics; they are deemed to be biased. As a result, some user groups enjoy privileges while others suffer disadvantages. They didn't explain anything about bias and effectiveness in the user biometrics while authenticating or generating cryptographic keys.

**3) *Inaccuracy*:** The fuzzy extractor relies too much on expert knowledge and needs more capacity to gain insight from data. Therefore, the *Gen(.)* and *Rep(.)* functions can still have the chance of false rejection/accepting a legitimate user for entering the system credentials. The scheme presented is fuzzy extractor-based, in which a user authentication of biometrics is performed many times (fuzzy extractor is a challenge-response verification method), which causes heavy computation and communication costs. Also, a legitimate user can easily trace through biometrics/facial recognition/thumb extraction.

**4) *Stolen-Verifier Attack*:** Suppose an attacker steals the smart card and uses power analysis or reverse engineering techniques, $\mathcal{A}$ chooses two random numbers $L_A$, $L_B$ computes $M_A = h(MID_i||L_A)$, $B_A = M_A \oplus h(L_A||L_B)$, obtain $\{B_A, L_A\}$ which means $\mathcal{A}$ can reaches the identity and password, and then $\mathcal{A}$ uses it malicious deeds like launching replay, DoS, masquerade, impersonation, ESL, MITM, side-channel and other attacks. Therefore, a stolen verifier attack is possible on the scheme [17].

Considering all the drawbacks mentioned above, it has been concluded that [17] is a weaker scheme.

**5) *Tracebality, DoS and Replay Attacks*:** In the login and authentication phase of the protocol, the first message transmitted from the mobile user towards the registration centre is M1 = $\{r_1^{/}, B_1, D_1, D_2, T_1\}$. This message contains server identity $SID_j$, which an attacker can easily capture/identify in $D_1 = SID_j \oplus h(TMID_i||T_1)$ and violate the system's privacy. Similarly, attacker can also use this server identity ($SID_j$) to launch replays and DoS attacks on the system. Therefore, the Wu et al. [17] scheme suffers from privacy issues and is vulnerable to traceability, DoS and Reply attacks.

**6) *Lack of Password Changing Phase*:** Despite using passwords by a user in the login and authentication phase of Wu et al.'s [17] scheme, they do not provide a facility for a legitimate user to change his/her password freely and securely.

All these vulnerabilities shall be mitigated by designing a robust, lightweight, and probable secure system for the teleworking environment.

## 3 System model

The proposed network model consists of a teleworking server (TS) that connects numerous Internet-of-Things (IoT) of the organization, a registration center (RC), and a remote-mobile user (MU). All the entities first register with the registration center (RC) over a reliable channel and then operationalize for the teleworkers in the teleworking environment. Suppose the RC is a fully trusted entity. In contrast, all other components, i.e., the Teleworking Server (TS) and mobile user (MU), may or may not be fully trusted, as shown in Fig 2. These participants can be described as follows:

• *Registration Center (RC)*: It is responsible for registering all the entities within the organization (Teleworking server) or outside of the organization (remote/mobile user).

• **Teleworking Server (TS)**: The teleworking server is placed between end-users and the IoT of the organization for data processing and broadcasting. Furthermore, TS provides reliable services with smaller latency to the remote user.

• *Mobile-User (MU)*: It is either PCs, tablets, cell phones, or other network-enabled devices to get facilities prescribed by the teleworking server (TS)

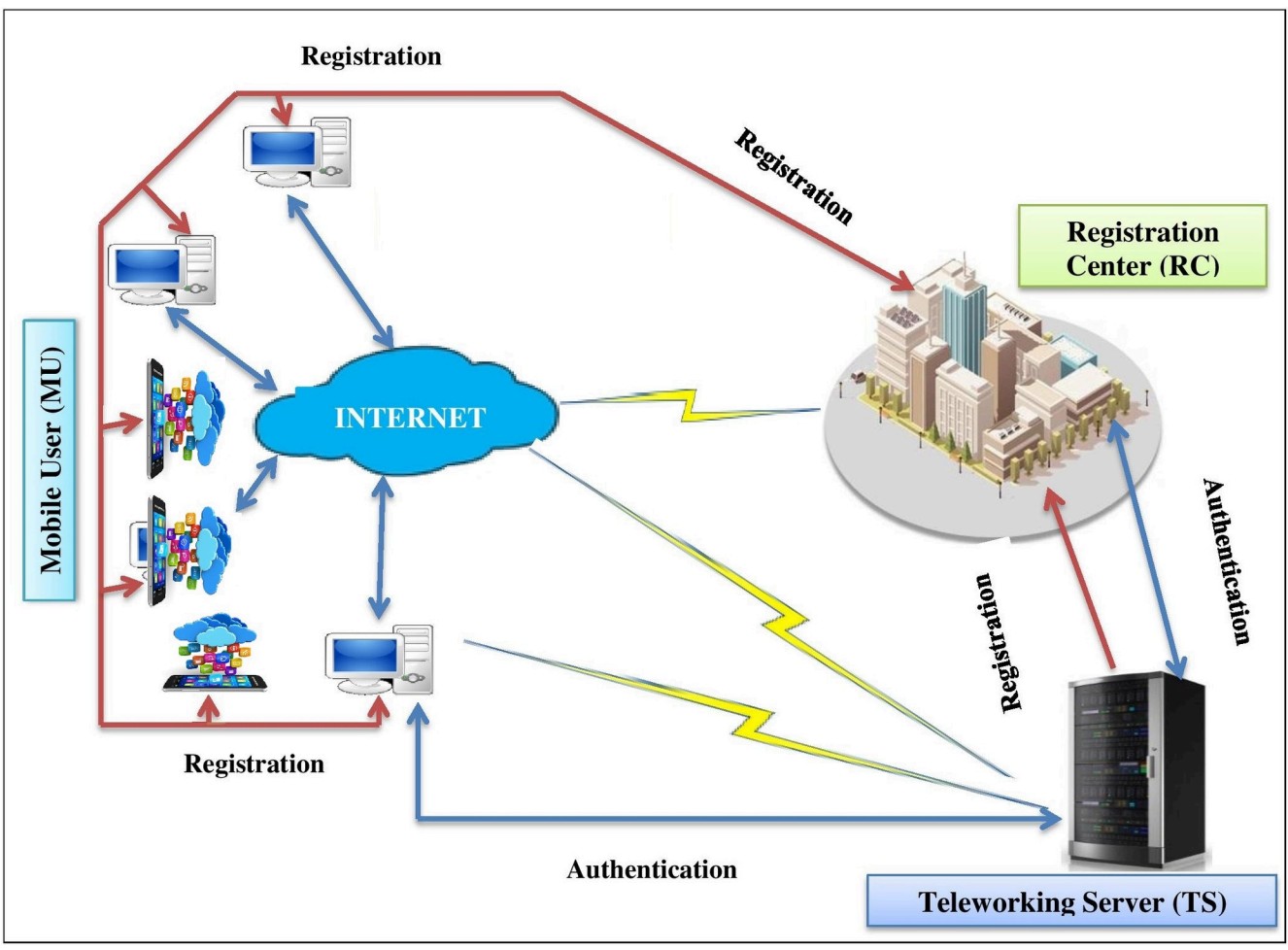

**Fig 2. System model.**

As discussed, the popularity of teleworking is growing daily, allowing people to work from anywhere, utilizing various mobile devices and Internet services. Teleworking boosts business efficiency, productivity, and expenses, but securely accessing information, operating Internet-of-Things (IoT) remotely, and maintaining privacy are challenging concerns that have yet to be resolved. As a result, this study developed a security method that can effectively alleviate the security and privacy risks associated with teleworking. We will focus on the following main points:

1. To design an architectural framework for a teleworker to utilize/access the resources remotely in the teleworking environment so that no one can weaken the remote access level.

2. To propose a security mechanism that resisted all known threats on the client side during teleworking and can guarantee secure communication.

3. To facilitate the teleworkers' secure communication for working without breaks, saving businesses' costs and managing time effectively.

4. To facilitate the skilled individual to work without worrying about hacking, data leaking, and fishing so that they can feel more flexibility while working in the teleworking environment.

5. The proposed scheme is without bilinear mapping, having no point multiplication exponentiation, and without symmetric encryption/decryption functions while offering excellent services to teleworkers.

6. The secret session key secrecy, confidentiality, and authorization have been verified formally using ROR/ProVerif and informally using illustrations. The result shows that the mechanism is robust and lightweight for such a vulnerable environment.

7. To comparatively analyze the designed scheme with present schemes regarding computation and communication overheads.

## 3.1 Design goals

The following goals can be achieved by designing a security mechanism [32,33] for teleworking environment protection:

- **G1:** Message authentication and integrity: mobile user/end-user (MU) mandatorily verifies the jurisdiction of the received message without being altered, modified, or forged by someone.

- **G2:** Confidentiality and Authorization: The request sent to the server or the response received by the end user must be confidential. No one can figure out its internal contents, and both peers must confirm the authenticity of each other and the message exchanged.

- **G3:** Conditional Privacy-Preserving: Except for the organization server, no other participants can trace the identity of the other participants.

- **G4:** Untraceability: Two sessions mandatorily will always start on a different key. Each session's key must differ from other sessions; otherwise, a malicious user can easily trace the legitimate user.

- **G5:** Physical Protection: Protection of the system from active and passive attacks means the system is protected physically.

- **G6:** Resilience to Insider Threat: The server is accessible from any storage table; anyone accessing the internal credentials must not extract helpful information from memory. The internally stored credentials will be available to the attacker in a non-readable format to avoid any future hazard to the system.

- **G7:** Mutual Authentication: Each peer must mutually authenticate before starting data broadcasting. Each participant can verify the legality of messages and identities from other entities. If the verification fails, there may be a forgery attack.

- **G8:** Perfect Forward Secrecy: The 160-bit long keys cannot compute session keys without knowing hash values. It means that the secrecy of the previous session is not forward secrecy affected, even if the $\mathcal{A}$ can identify the long-term secret key, but still, A cannot succeed for hashed and encrypted values. Therefore, the proposed key agreement protocol satisfies the perfect feature.

- **G9:** Resists Man-in-the-Middle Attack: The security mechanism must be able to detect intruders. Each round trip must contain a random check to avoid a man-in-the-middle attack.

- **G10:** Resists Denial-of-Service (DoS) Attack: Message freshness, randomization, and predetermined time threshold can deny any reply attack or DoS attack.

### 3.2 Threat model

This work adopted the threat model Dolev-Yao called DY-model [34]. According to this model, an attacker has the following capabilities:

- Attacker $\mathcal{A}$ can easily capture messages from the public network channel.

- Attacker $\mathcal{A}$ can modify the recorded message.

- Attacker $\mathcal{A}$ can delete the full or some part of a message captured from an open network channel.

- The attacker $\mathcal{A}$ can easily launch a reply attack.

- Attacker $\mathcal{A}$ can also divert the route of a message.

- Attacker $\mathcal{A}$ can guess different credentials from a publically transmitted message.

- An attacker $\mathcal{A}$ can steal the mobile device and obtain useful credentials using reverse engineering techniques.

- Attacker $\mathcal{A}$ might be a privileged insider sitting on the system.

## 4 Proposed protocol

The proposed protocol consists of registration, authentication, and password change phases. Each of these phases is described one by one as under, while the different notations used for designing the protocol are shown in Table 1.

### 4.1 Teleworking Server (TS) registration phase

This phase is accomplished in the following steps:

**TS1:** The teleworking server first generates $ID_{TS}$, and sends them towards the registration center.

**TS2:** The registration center (RC) generates its secret key $s$, random numbers $r_{TS}$, $x_{TS}$, computes $A = h(ID_{TS}||r_{TS})$, $B = h(ID_{TS}||s)$, $C = h(ID_{TS}||x_{TS})$, $D = r_{TS} \oplus C$, store $\{A, D, \text{and } x_{TS}\}$, and sends ($r_{TS}$, B) back towards teleworking server over a private channel.

**TS3:** The teleworking, upon receiving ($r_{TS}$, B), also stores it in its database, as shown in Fig 3.

### 4.2 Mobile User (MU) registration phase

This phase is completed in the following steps:

**Table 1. Notations and their descriptions.**

| Notation | Meaning | Notation | Meaning |
|---|---|---|---|
| RC | Registration Center | $s$ | RC secret number |
| TS | Teleworking Server | $r_1, r_2, r_3$ | Random numbers |
| MU | Mobile User | T | Time Threshold |
| $ID_{MU}$ | Identity of the mobile user | $T_c$ | Current timestamp |
| $ID_{TS}$ | Identity of the teleworking server | $\mathcal{A}$ | Adversary or Attacker |
| SK | Session secret key | $\oplus$ | XOR function |
| $\Delta T$ | Maximum transmission delay | $r_{TS}$ | TS random number |
| $x_{TS}$ | Secret values of RC | $||$ | Concatenation function |
| LA1 to LA5 | Login and Authentication Step 1 to 5 | G1 to G10 | Goal1 to Goal 10 |

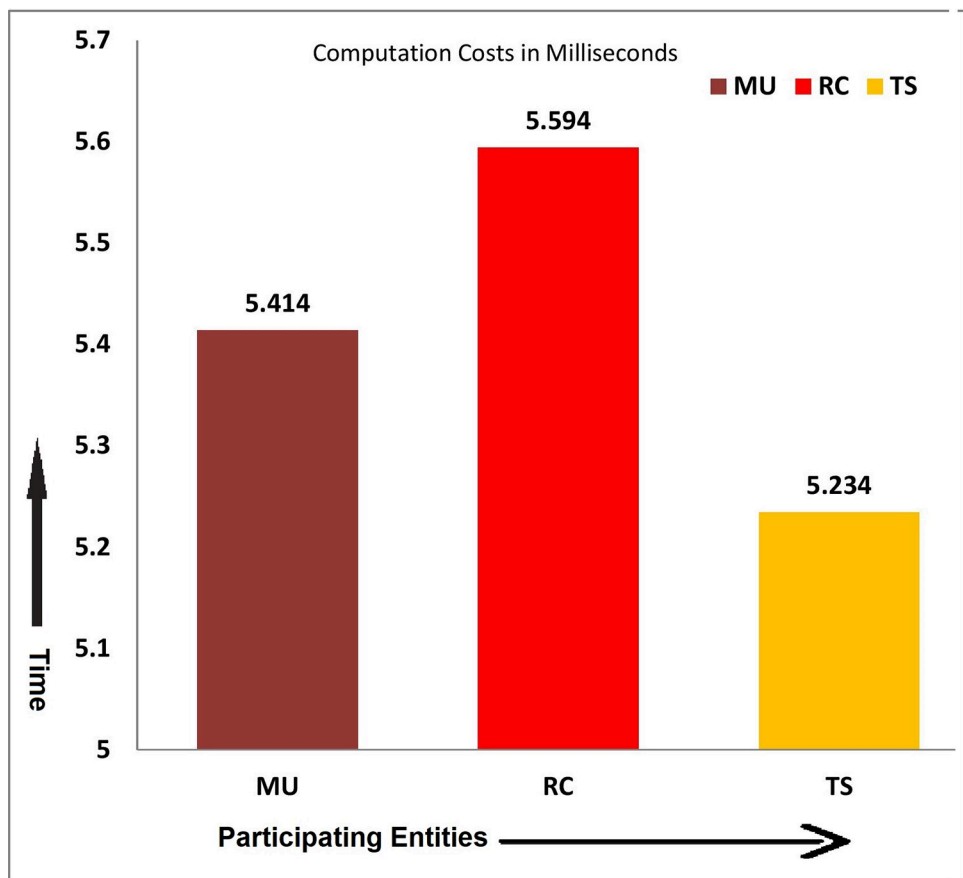

**Fig 3. Teleworking Sever (TS) registration phase.**

**MU1:** The mobile user generates its identity $ID_{MU}$ and sends it toward the registration center over a secure path.

**MU2:** The registration center (RC) first chooses its secret values $s$ random numbers $r_{MU}$, $x_{MU}$, computes $A' = h(ID_{MU}||r_{MU})$, $B' = A' \oplus h(s||x_{MU})$, stores $\{A', B', h(.)\}$ and transmits $(A', B')$ back towards the mobile user (MU) over a private channel.

**MU3:** When receiving $(A', B')$, the mobile user also stores it in its memory, as shown in Fig 4.

## 4.3 Login and authentication phase

The protocol's most essential and logical phase is the login and authentication phase. This phase is completed in the following steps.

**LA1:** The mobile user (MU), first, provides their identity $ID_{MU}$, password PW, computes $D = (A'||B') \oplus h(ID_{MU}||PW||r_1)$, $E = ID_{MU} \oplus h(D||PW||r_1)$, $r_1' = r_1 \oplus h(ID_{TS}||ID_{MU})$, record $T_1$, compute: $F = ID_{TS} \oplus h(ID_{MU}||T_1)$, $G = h(ID_{TS}||ID_{MU}||r_1||T_1)$, and transmits $(B', r_1', F, G, T_1)$ towards registration center (RC) over an open network channel.

**LA2:** The RC, first checks the time interval with the maximum available time threshold $T_C$-$T_1 \le \Delta T$; if it doesn't validate, the potential reply attack is considered; otherwise, check $B'? = B' = A' \oplus h(s||x_{MU})$, it doesn't match, the process will be denied, else compute: $J = F \oplus h(ID_{MU}||T_1)$, $r_1' = r_1 \oplus h(ID_{TS}||ID_{MU})$, $G' = h(ID_{TS}||ID_{MU}||r_1||T_1)$, and confirms $G? = G'$, if becomes valid, the onward computation performed, otherwise, the process is terminated for potential DoS attack. The RC now retrieves A and selects $T_2$ and $r_2$. Computes $Q_1 = h(A||r_2) \oplus h(h$

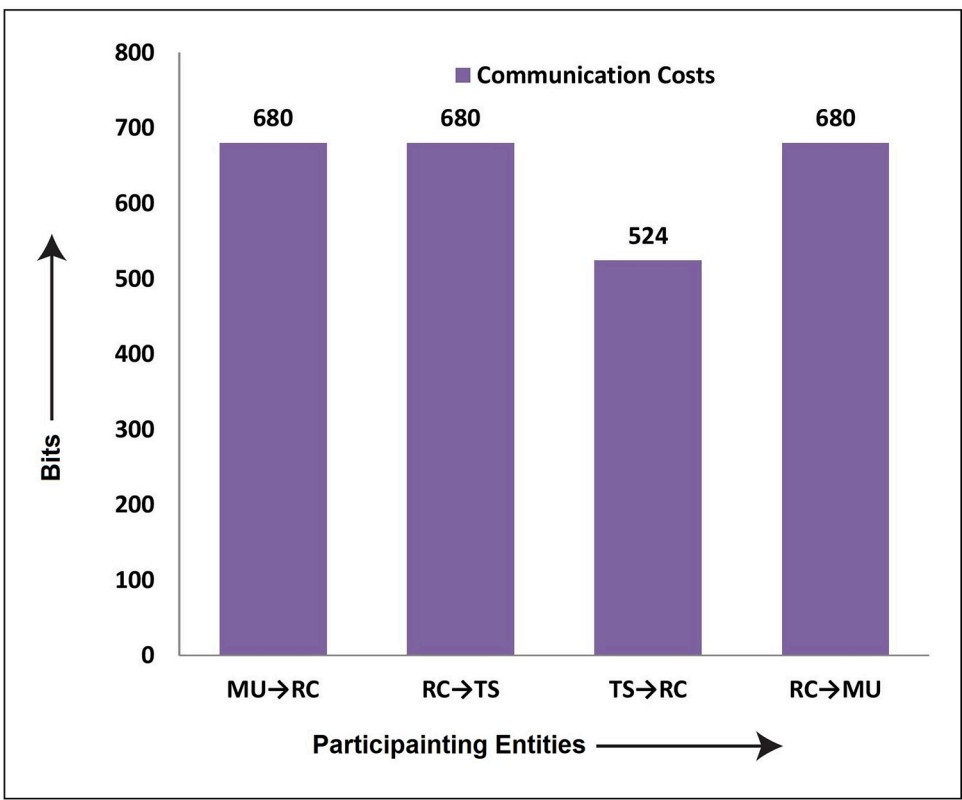

**Fig 4. Mobile User (MU) registration phase.**

$(ID_{TS}||s))$, $Q_2 = h(ID_{TS}||x_{TS})$, $r_2' = D \oplus Q_2$, $A'' = h(A||r_2') \oplus A'$, $Q_3 = h(h(A||r_2||ID_{TS}||T_2)$, and transmits $(r_1', A'', Q_2, Q_3, T_2)$ towards teleworking server (TS) over the open path.

**LA3:** The TS first checks the time interval with the maximum available time threshold $T_C$-$T_1 \leq \Delta T$; if it doesn't validate, the potential reply attack is considered; otherwise, computes $h(A||r_2) = Q_1 \oplus h(B)$, $Q_3' = h(h(A||r_2||ID_{TS}||T_2)$, checks $Q_3$? = $Q_3'$, if validated, the TS selects $T_3$, $r_3$ and computes $A = h(ID_{TS}||x_{TS})$, $A' = A'' \oplus h(ID_{TS}||x_{TS})$, $r_1' = r_1 \oplus h(ID_{TS}||A)$, $SK = h(h(A||r_2||r_1||r_3||A')$, $r_3' = r_3 \oplus h(r_1||ID_{TS})$, $Q_4 = h(r_1||r_3||T_3)$, $Q_5 = h(SK||h(A||r_2)||r_1)$, and broadcast $(r_3', Q_4, Q_5, T_3)$ back towards RC, over an insecure channel.

**LA4:** The RC, first checks the time interval with the maximum available time threshold $T_C$-$T_3 \leq \Delta T$; if it doesn't validate, the potential reply attack is considered; otherwise, record timestamp $T_4$, computes $Q_4 = h(r_1||r_3||T_3)$, confirm $Q_4$? = $Q_4'$, if found valid, computes $r_3' = r_3 \oplus h(r_1||ID_{TS})$, $SK = h(h(A||r_2||r_1||r_3||A')$, $Q_5' = h(SK||h(A||r_2)||r_1)$, check $Q_5'$? = $Q_5$, compute: $Q_6 = h(A||r_2) \oplus h(r_1||A')$, $Q_7 = h(h(A||r_2)||r_3||T_4)$ and sends $(r_3', Q_5, Q_6, Q_7, T_4)$ to end-user via open channel.

**LA5:** The MU, first checks the time interval with the maximum available time threshold $T_C$-$T_4 \leq \Delta T$; if it doesn't validate, the potential reply attack is considered; otherwise, computes $r_3' = r_3 \oplus h(r_1||ID_{TS})$, $h(A||r_2) = Q_6 \oplus h(r_1||A)$, $Q_7' = h(h(A||r_2)||r_3||T_4)$, confirms $Q_7$? = $Q_7'$, if fund valid, calculates $SK = h(h(A||r_2||r_1||r_3||A')$, $Q_5 = h(SK||h(A||r_2)||r_1)$, checks $Q_5$? = $Q_5'$, if validates, keeps it the secret session key for future communication as shown in Fig 5.

## 4.4 Password change phase

If the mobile user (MU) desires to change their password, the proposed protocol offers the facility of changing it securely. In this regard, MU enters their identity $ID_{MU}$, password PW and calculates: $A^m = h(ID_{MU}||PW) \oplus A$, $A'^m = h(ID_{MU}||PW) \oplus A^m$, chooses a random number

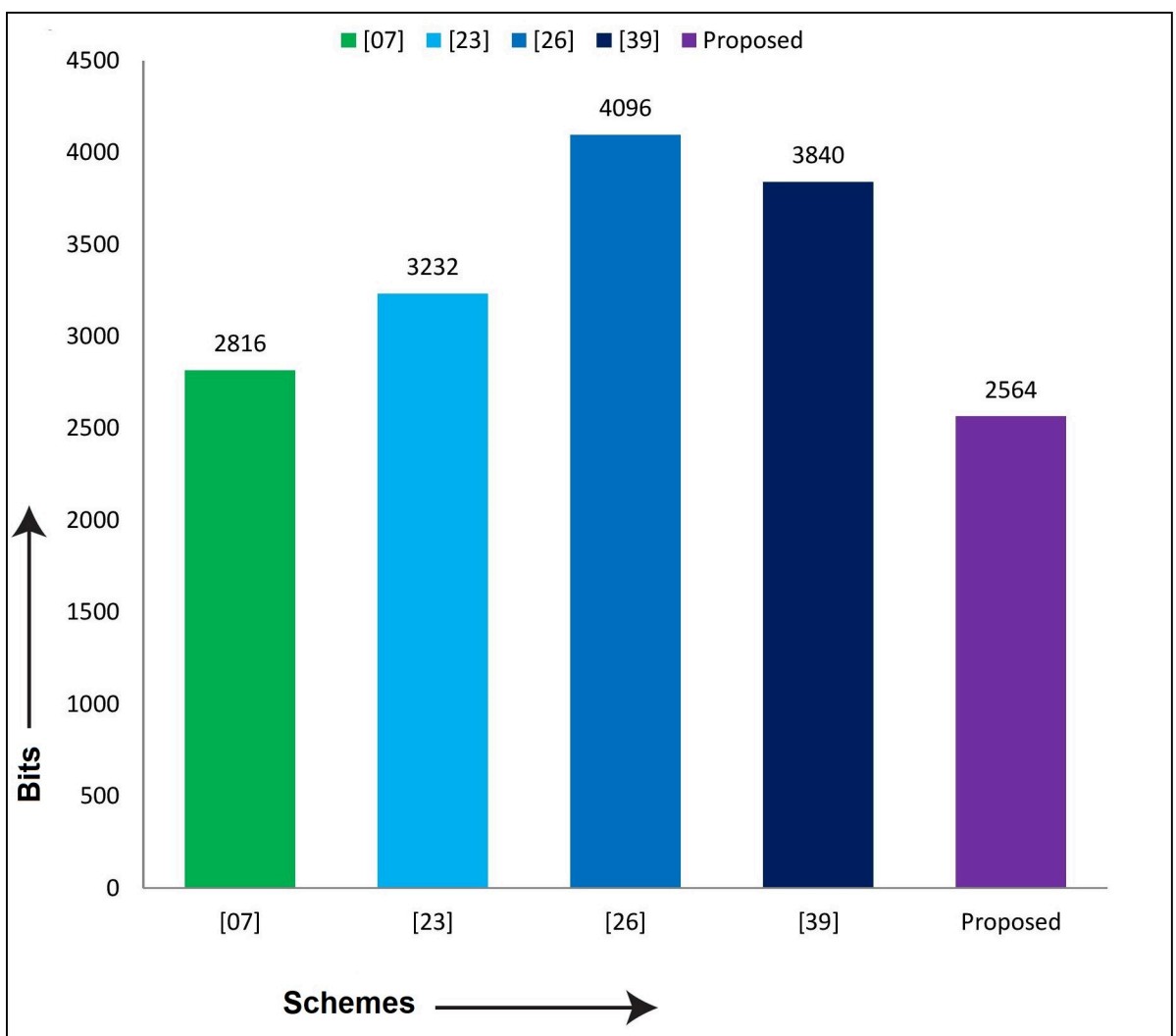

**Fig 5. Login & authentication phase. Remark:** The clock synchronization issue can be addressed by configuring each participant to the global clock so that it will establish the start and finish time slot as well as correct the offset and drift rate of the participants' clock w.r.t global time.

$r_5 \in Z^*_n$ and calculates: $A^{/m} = h(A^{/m}||r_5) \oplus A^m$. If $A^{/m}? = A^m$, asked the MU to enter a new password $PW^{new}$, upon provision of the new password $PW^{new}$, locally computes: $A = h(ID_{MU}||s)$, $A^{/} = h(PW^{new}||r_5)$, $A^{new} = h(ID_{MU}||PW^{new}) \oplus A$, $A^{/new} = h(ID_{MU}||PW^{new}) \oplus A^{new}$ and replaces $\{A^m, A^{/m}\}$ with $\{A^{new}, A^{/new}\}$, as shown in Fig 6.

## 5 Security analysis

In this section, the security of the proposed protocol can be analyzed using the random oracle model (ROM) [35] and ProVerif2.03 [36], which is also used by [18,21,26,28,30]. These are described one by one as follows:

### 5.1 ROM analysis

A standard formal security analysis method, namely the ROM, is used to analyse the proposed protocol's shared session key between MU and RC and then RC and TS against an adversary

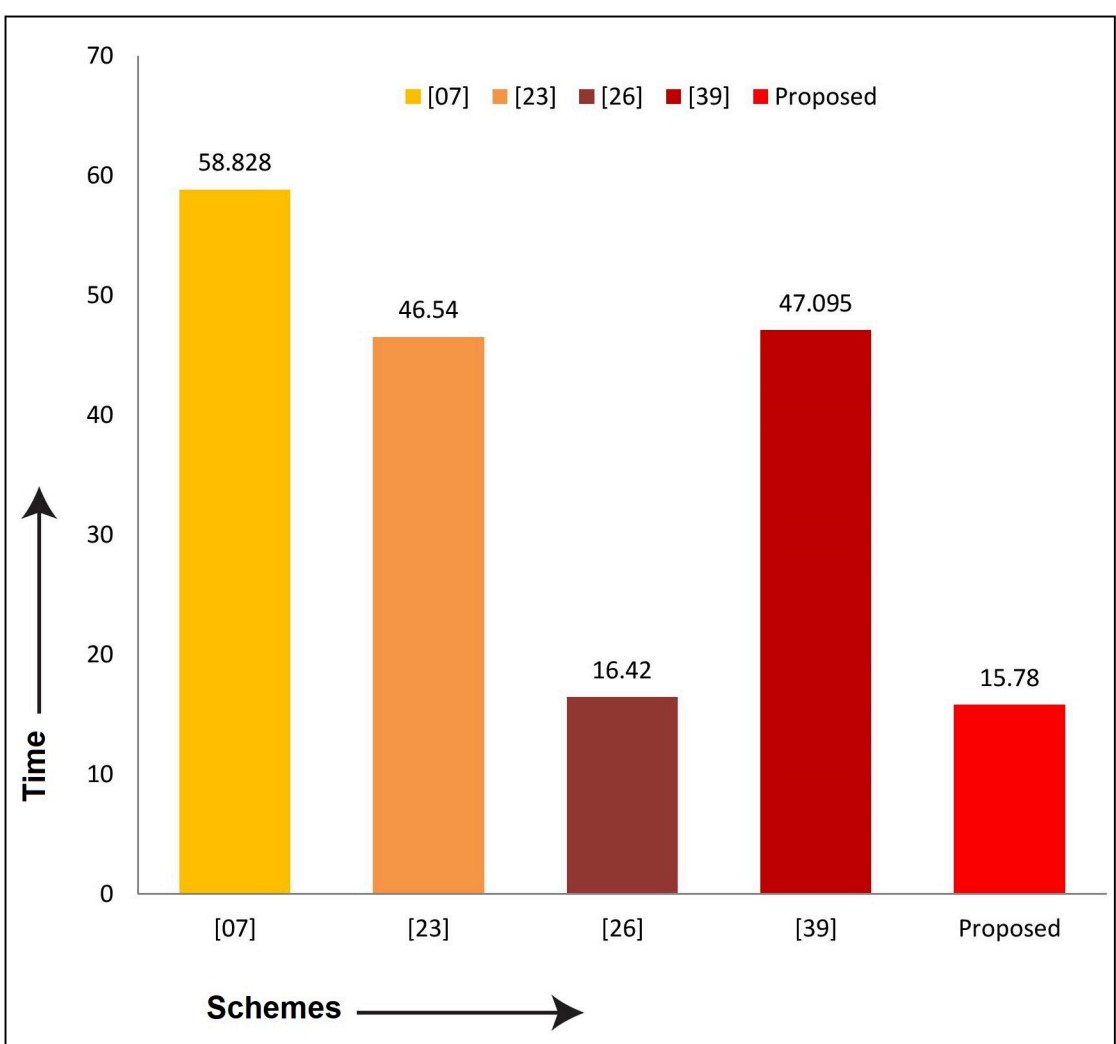

**Fig 6. Password change phase.**

$\mathcal{A}$. For achieving this goal, we first will go for the semantic approach and session key security. The different queries executed by an adversary $\mathcal{A}$ are discussed below; however, the collision-resistant one-way hash function will also be one of the participants for $\mathcal{A}$, which can be demonstrated as H A S H, so by keeping these, the ROR model performed numerous elements, including RC, TS, MU in which MU and TS are engaged for mutual authentication apart from RC which primarily involved in the registration phase of the protocol. Let $\prod_{MU}^{b_1}$ and $\prod_{TS}^{b_2}$ identify the $b_1$ and $b_2$ instances of MU and TS correspondingly, also called random oracle instances.

1) ***Execute*** ($\prod_{MU}^{b_1}$, $\prod_{TS}^{b_2}$): $\mathcal{A}$ eavesdrop on their own message among the shared message between MU and TS using this query.

2) ***Corrupt*** ($\prod_{MU}^{b_1}$): $\mathcal{A}$ forge parameters from the memory of a compromised MU using this query.

3) ***Reveal($\Pi^b$)***: $\mathcal{A}$ can disclose the secret session key SK between MU and RC, RC, MU and other participants using this query.

**4) Test($\Pi^b$):** $\mathcal{A}$ can test by calling $\Pi^b$ to check the originality of SK, and $\Pi^b$ received should be random for $\mathcal{A}$, and $\mathcal{A}$ will definitely be flipped a coin, say $d$ (Accepted Instance); in such a case, the following three cases will happen.

- $\prod_{MU}^{b_1}$ and $\prod_{TS}^{b_2}$ accepted state
- $\prod_{MU}^{b_1}$ and $\prod_{TS}^{b_2}$ session identification state
- $\prod_{MU}^{b_1}$ and $\prod_{TS}^{b_2}$ mutual participation state

**5) Semantic Security:** Let $\mathcal{A}$ be fully authorized to run the Test(.) query and try to interfere P by polynomial tries. Individually, the three algorithms, i.e., *Execute(.)*, *Send(.)*, and *Hash(.)*, are run in $q_E$, $q_S$, and $q_H$. The *Test*(.) query once at most. Let $l_h$ be the length in a bit for a collision-resistant one-way hash function, and $n = 2^{l_h}$ be the average length of hash operation for another transcript in P. Then the advantage with $\mathcal{A}$ in breaking P by polynomial times attempt can be expressed as:

$$Adv_{\mathcal{A}}^{\mathcal{P}} \leq \frac{(q_S + q_E)^2}{n} + \frac{q_H^2}{2^{l_h}} + 2Adv_{\mathcal{A}}^{\text{SHA1}}$$

To pretend the different attacks on P, we justify through different games. The event $Sus_{\mathcal{A}}^i (0 < i < 3)$ matching to these games means that $\mathcal{A}$ completes its goal in that specific game, which is defined one by one as under:

**Game 0:** In this environment, $\mathcal{A}$ launch a genuine attack on P. To do so, the probability with $\mathcal{A}$ in cracking P is represented in Eq (1).

$$Adv_{\mathcal{A}}^{\mathcal{P}} = |2\Pr[Sus_{\mathcal{A}}^0] - 1| \tag{1}$$

**Game 1:** In this environment, $\mathcal{A}$ launches *Execute(.)* and *Test(.)* queries for verifying the obtained results according to protocol's P transcripts ($B'$, $r_1'$, F, G, $T_1$) which is related to session secret key SK. Conversely, due to random numbers for each round trip, $\mathcal{A}$ doesn't diagnose the relationship of ($B'$, $r_1'$, F, G, $T_1$) with their obtained result through *Test(.)* query. However, the probability with $\mathcal{A}$ in identifying the relationship of P's transcripts is represented in Eq (2).

$$\Pr[Sus_{\mathcal{A}}^1] = \Pr[Sus_{\mathcal{A}}^0] \tag{2}$$

**Game 2:** Here $\mathcal{A}$ calculates the session secret key SK from the messages transmitted over a public network channel, i.e. SK = h(h(A||$r_2$||$r_1$||$r_3$||$A'$), but due to using SHA1 of key size 156-bits, difficult for $\mathcal{A}$; however, the probability with $\mathcal{A}$ in computing SK from the openly transmitted messages of P is represented in Eq (3).

$$\Pr[Sus_{\mathcal{A}}^2] - \Pr[Sus_{\mathcal{A}}^1] = Adv_{\mathcal{A}}^{\text{SHA1}} \tag{3}$$

**Game 3:** In this environment, $\mathcal{A}$ runs *Execute(.)* and *Send(.)* queries to launch a hash image collision attack. According to the birthday paradox [32], the risk of hash collision is $\frac{q_H^2}{2^{l_h+1}}$. Meanwhile, the collision probability of other transcripts is $\frac{(q_S+q_E)^2}{2n}$. Therefore, the probability with $\mathcal{A}$ for hash collision in P is represented in Eq (4).

$$\Pr[Sus_{\mathcal{A}}^3] - \Pr[Sus_{\mathcal{A}}^2] \leq \frac{(q_S + q_E)^2}{2n} + \frac{q_H^2}{2^{l_h+1}} \tag{4}$$

Similarly, in the random bit $r \in (0, 1)$, the probability with $\mathcal{A}$ of guessing the random number

of P is represented in Eq (4).

$$\Pr[Sus^3_{\mathcal{A}}] = \frac{1}{2} \tag{5}$$

Now, combine Eqs (1)–(5), we get

$$\frac{1}{2} Adv^{\mathcal{P}}_{\mathcal{A}} \leq \frac{(q_S + q_E)^2}{2n} + \frac{q_H^2}{2^{l_h+1}} + Adv^{\mathrm{SHA1}}_{\mathcal{A}} \tag{6}$$

Eq (6) can also be expressed as:

$$Adv^{\mathcal{P}}_{\mathcal{A}} \leq \frac{(q_S + q_E)^2}{n} + \frac{q_H^2}{2^{l_h}} + 2Adv^{\mathrm{SHA1}}_{\mathcal{A}}$$

## 5.2 Security model

This model consists of two peers–the Adversary $\mathcal{A}$ and–the Responder Ř. $\mathcal{A}$ communicate either mobile user (MU) or TS or RC, but we denote all of them a Ë$^\sigma$ which means $\sigma^{\mathrm{th}}$ instance of any of them either MU, TS or RC. $\mathcal{A}$ launches the queries, and responses received from Responder Ř are shown in Table 2.

- Ë$^{\mathrm{Sn}}$ means $\mathcal{A}$ impersonate MU, TS, RC by forcing $r_1$ or $r_2$

- Ë$^{\mathrm{Ns}}$ means $\mathcal{A}$ forges $r_3$ or $r_4$ from the participants

- Ë$_{\mathrm{SS}}$ means $\mathcal{A}$ overcomes the semantic security of the protocol.

## 5.3 ProVerif2.03 simulation

This is a formal security analysis method in which we will check the key's robustness, key's secrecy, confidentiality, and reachability; a programming verification toolkit ProVerif2.03 [36] is in S1 Appendix of this article. However, upon running the code, the result shows that the attacker couldn't crack the secret session key at any computation stage. The status of SK is secure, and its confidentiality and reachability are preserved as shown under:

————————————————

*Verification summary:*

*Query not attacker(SK[]) is true.*

*Query inj-event(end_MU(IDmu[])) = = > inj-event(start_MU(IDmu[])) is true.*

*Query inj-event(end_RC(IDrc[])) = = > inj-event(start_RC(IDrc[])) is true.*

*Query inj-event(end_TS(IDts[])) = = > inj-event(start_TS(IDts[])) is true.*

————————————————

**Table 2. Queries and their response.**

| |
|---|
| **Setup**: By running this query, the challenger $\mathbb{C}$ returns the obtained parameters to $\mathcal{A}$. |
| **$h(Message_k)$**: The challenger $\mathbb{C}$ stores a $L_{hx}$, querying $h(Message_k)$, extract $r_k \in Z^*_p$ and record the result {$Message_k$, $r_k$} in $L_{hx}$; if $Message_k$ is not found, again stores the result in $L_{hx}$ and return $r_k$ to $\mathcal{A}$. |
| **$MAC(k, Message_k)$**: The stored list $L_{Mx}$ with $\mathbb{C}$ comprising of different tuples in the form of MAC(k, $Message_k$, M), $\mathbb{C}$ querying MAC(k, $m_k$), extract $M \in Z^*_p$ and store {k, $m_k$, M} in $L_{Mx}$, if not found, return M to $\mathcal{A}$. |
| **$Send(\text{Ë}^\sigma, M_\sigma)$**: The challenger $\mathbb{C}$ sends this message towards the proposed authentication protocol and communicates the output received with $\mathcal{A}$. |
| **$Execute(MU, RC)$**: The result obtained by $\mathbb{C}$ while using his query is $r_1$, $r_2$ and shared with $\mathcal{A}$. |
| **$Execute(RC, TS)$**: The result from this query $r_3$, $r_4$ is shared with $\mathcal{A}$. |
| **$Reveal(\text{Ë}^\sigma)$**: The challenger $\mathbb{C}$ yields the present secret session key SK with Ë$^\sigma$ and $\mathcal{A}$. |
| **$Test(\text{Ë}^\sigma)$**: $\mathcal{A}$ demands Ë$^\sigma$ for SK; the Ë$^\sigma$ communicate with $\mathcal{A}$ but $\mathcal{A}$ flipping a coin 1-win, 0-lose. |

### 5.4 GNY logic analysis

The GNY logic [37] is a formal method of security analysis which Gong-Needham-Yahalom first introduced for the formal proof of a security protocol. The different formulae and statements used in this logic are shown in Table 3.

Now, we are using GNY logic for the proposed protocol and make assumptions which are as follows:

$$MU \ni \#(r_1) \tag{7}$$

$$MU \ni \#r_1^{/} \tag{8}$$

$$RC \ni \#s \tag{9}$$

$$RC \ni \#x_{MU} \tag{10}$$

$$RC \ni \#x_{TS} \tag{11}$$

$$RC \ni \#(r_1) \tag{12}$$

$$RC \ni \#(r_2) \tag{13}$$

$$RC \ni \#(r_3) \tag{14}$$

$$RC \ni \#r_2^{/} \tag{15}$$

$$TS \ni \#x_{TS} \tag{16}$$

$$TS \ni \#(r_1) \tag{17}$$

$$TS \ni \#(r_2) \tag{18}$$

$$TS \ni \#(r_3) \tag{19}$$

**Table 3. Formulas and statements used in GNY logic.**

| Formula/Statement | Description |
|---|---|
| (A, B) | Combining A with B |
| h(A) | Hashing of A |
| *A:A | A is not initiated here |
| P◁A | P sees A |
| P∋A | P owns A |
| P\|∼A | P taken A |
| P\|≡#(A) | P believes the freshness of A |
| P\|≡φ(A) | P recognize A |
| P\|≡ASK↔$^{S_K}$B | A believes $S_K$ is the secret session key among A and B |
| P\|⇒A | P control A |
| P◁*A | P sees A, which hasn't been delivered previously for the current session |

$$TS \ni \#r_3^{/} \tag{20}$$

We transformed the proposed scheme to P→Q: (X) from fill-in GNY logic and made some changes to notations.

Using GNY logic, the proposed scheme can be represented as:

$$MU \rightarrow RC : (B^{/}, r_1^{/}, F, G, T_1) \tag{21}$$

$$MU \rightarrow (A^{/} \oplus h(s||\mathrm{x}_{MU})) \tag{22}$$

$$MU \rightarrow ((h(ID_{MU}||(\mathrm{r}_{MU}))) \oplus h(s||\mathrm{x}_{MU}))) \tag{23}$$

$$MU \rightarrow (ID_{TS} \oplus h(ID_{MU}||\mathrm{T}_1)) \tag{24}$$

$$MU \rightarrow (h(ID_{TS}||ID_{MU}||\mathrm{r}_1||\mathrm{T}_1)) \tag{25}$$

$$RC \rightarrow TS : (r_1^{/}, A^{//}, \mathrm{Q}_2, \mathrm{Q}_3, \mathrm{T}_2) \tag{26}$$

$$RC \rightarrow (h(A||r_2^{//}) \oplus A^1) \tag{27}$$

$$RC \rightarrow (h(ID_{TS}||\mathrm{x}_{TS})) \tag{28}$$

$$RC \rightarrow (h(h(A||\mathrm{r}_2||ID_{TS}||\mathrm{T}_2)) \tag{29}$$

$$TS \rightarrow RC : (r_3^{/}, \mathrm{Q}_4, \mathrm{Q}_5, \mathrm{T}_3) \tag{30}$$

$$RC \rightarrow (h(\mathrm{r}_1||\mathrm{r}_3||\mathrm{T}_3)) \tag{31}$$

$$RC \rightarrow (h(SK||h(A||\mathrm{r}_2)||\mathrm{r}_1)) \tag{32}$$

$$RC \rightarrow MU : (r_3^{/}, \mathrm{Q}_5, \mathrm{Q}_6, \mathrm{Q}_7, \mathrm{T}_4) \tag{33}$$

$$RC \rightarrow (h(SK||h(A||\mathrm{r}_2)||\mathrm{r}_1)) \tag{34}$$

$$RC \rightarrow (h(A||\mathrm{r}_2) \oplus h(r_1||A^{/}) \tag{35}$$

$$RC \rightarrow (h(h(A||\mathrm{r}_2)||\mathrm{r}_3||\mathrm{T}_4)) \tag{36}$$

The mobile user keeps $ID_{MU}$, and $PW_{MU}$ in its memory and can extract $\mathrm{r}_1$ during computations, so by applying the GNY logic and Eq (7), we have

$$\frac{MU \ni ID_{MU}, MU \ni PW_{MU}, MU \ni \mathrm{r}_1}{MU \ni D} \tag{37}$$

Eqs (12) and (37), we have

$$\frac{RC \triangleleft D}{RC \ni D} \tag{38}$$

Eqs (17) and (38), we have

$$\frac{TS \ni ID_{TS}, TS \ni PW_{TS}, TS \ni r_1}{MU \ni J} \tag{39}$$

$$\frac{TS \triangleleft J}{TS \ni J} \tag{40}$$

Eqs (37) and (39) become:

$$\frac{RC \ni ID_{MU}, RC \ni r_1}{RC \ni ID_{MU}} \tag{41}$$

Eq (41) becomes:

$$\frac{RC \ni ID_{MU}, RC \ni SK, RC \ni F, RC \ni r_1}{RC \ni G} \tag{42}$$

$$\frac{RC \ni E, RC \ni B^/}{RC \ni B^1} \tag{43}$$

Keeping Eqs (42) and (43), and the credentials checking by mobile-user in which each part/parameter of the message passed/verified to and from RC, as shown as under:

$$\frac{MU \ni r_1, MU \ni ID_{MU}, MU \ni D}{MU \ni J} \tag{44}$$

$$\frac{MU \ni D, MU \ni J}{MU \ni Q_3^/} \tag{45}$$

$$\frac{MU \ni r_2, MU \ni ID_{TS}, MU \ni E}{MU \ni A^/} \tag{46}$$

$$\frac{MU \ni G^/, MU \ni r_1'}{MU \ni Q_3^/} \tag{47}$$

Keeping Eqs (46) and (47), and the credentials checking by teleworking-server in which each part/parameter of the message passed/verified to and from RC, as shown under:

$$\frac{TS \ni ID_{TS}, TS \ni D, TS \ni r_1}{u \ni Q_3^/} \tag{48}$$

$$\frac{TS \ni ID_{TS}, TS \ni r_2, TS \ni J}{TS \ni A^{//}} \tag{49}$$

$$\frac{TS \ni ID_{MU}, TS \ni r_3, TS \ni Q_3}{TS \ni SK} \tag{50}$$

$$\frac{TS \ni ID_{MU}, TS \ni J, TS \ni Q_3, TS \ni J, TS \ni T_1}{TS \ni Q_4} \tag{51}$$

From Eqs (44) and (45), we have:

$$\frac{\mathrm{TS} \ni ID_{MU}, \mathrm{TS} \ni \mathrm{J}, \mathrm{TS} \ni r_3, \mathrm{R} \ni Q_3}{\mathrm{TS} \ni \mathrm{SK}} \tag{52}$$

$$\frac{\mathrm{TS}| \equiv \#(r_1), \mathrm{TS}| \equiv \#(ID_{MU}), \mathrm{TS}| \equiv \#(\mathrm{J})}{\mathrm{TS}| \equiv \#Q_3} \tag{53} \quad \textbf{Goal1Achieved}$$

From Eqs (48)-to-(51), the same for RC, as RC has full control over the message in, $(B^/, r_1^/, F, G, T_1)$ and out $(r_1^/, A^{//}, Q_2, Q_3, T_2)$, so we have

$$\frac{RC \ni ID_{TS}, RC \ni \mathrm{SK}, RC \ni \mathrm{J}, RC \ni r_1}{RC \ni \mathrm{J}} \tag{54}$$

$$\frac{RC| \equiv \#(ID_{TS}), RC| \equiv \#(A_1^{//}), RC| \equiv \#Q_3}{RC| \equiv \#r_3^/} \tag{55}$$

$$\frac{RC| \equiv \#(Q_3^/), RC| \equiv \#(Q_2), RC| \equiv \#(T_1)}{RC \ni \mathrm{J}} \tag{56} \quad \textbf{Goal2Achieved}$$

From Eqs (52)-to-(56), again for RC because the RC has complete control over the message in, $(B^/, r_1^/, F, G, T_1)$ and out $(r_1^/, A^{//}, Q_2, Q_3, T_2)$, so we have

$$\frac{RC \ni \mathrm{ID}_{MU}, RC \ni \mathrm{SK}, RC \ni Q_3, RC \ni r_2, RC \ni T_1}{RC \ni r_3^/} \tag{57}$$

$$\frac{RC \ni Q_5, RC \ni Q_6, RC \ni Q_4^/, RC \ni r_1^/, RC \ni T_2}{RC \ni \mathrm{SK}} \tag{58}$$

$$\frac{RC \ni \mathrm{ID}_{TS}, RC \ni ID_{TS}, RC \ni \mathrm{SK}, RC \ni Q_5, RC \ni x, RC \ni r_1, RC \ni r_2, RC \ni Q_5^/}{RC \ni \mathrm{SK}} \tag{59}$$

$$\frac{RC \ni r_3, RC \ni s, RC \ni x_{MU}, RC \ni T_3}{RC \ni Q_7^/} \tag{60}$$

$$\frac{RC \ni A_1^{//}, RC \ni Q_2, RC \ni \mathrm{ID}_{MU}, RC \ni ID_{TS}}{RC \ni A^{//}} \tag{61}$$

$$\frac{RC \ni ID_{TS}, \mathrm{PCS} \ni x_{TS}, RC \ni r_2^/, RC \ni Q_2, RC \ni A_2^{//}}{RC \ni ID_{TS}} \tag{62}$$

Like RC (Eqs (57)-to-(62)), we will use the GNY logic for a mobile user, as MU also sees

$(r_3^/, Q_4, Q_5, T_3)$ and $(r_3^/, Q_5, Q_6, Q_7, T_4)$, messages, so, we have :

$$\frac{MU \ni r_3, MU \ni s, MU \ni x_{MU}, MU \ni T_3}{RC \ni Q_7^/} \tag{63}$$

$$\frac{MU \ni A_1^{//}, MU \ni Q_2, MU \ni Q_3, MU \ni x, M \ni T_4}{MU \ni SK} \tag{64}$$

$$\frac{MU \ni A_1^{//}, MU \ni Q_2^/, MU \ni ID_{MU}, MU \ni ID_{TS}}{MU \ni A_1^{//}} \tag{65}$$

$$\frac{MU| \equiv \#(r_1), MU| \equiv \#(s), MU| \equiv \#(r_3), MU| \equiv \#(T_4)}{MU| \equiv \#Q_5} \tag{66} \quad \textbf{Goal3Achieved}$$

## 5.5 Informal security analysis

Assume an adversary $\mathcal{A}$ has complete control over the open network and can intercept, manipulate, delete, or update the communication transmission between participants. Then, there's how the presented authentication system will withstand identified weaknesses. In this section of the study, we shall address such assumptions one by one as follows:

**5.5.1 Resists side channel attack.** The presented security mechanism is generally less dependent on key numbers, strongly validates the main attributes at various steps, and calculates a different secret session key for each session, which leads to the sequence of operations changing for another session. Similarly, the availability of timestamps at each round trip of the protocol and the exchange of random numbers differently for the different sessions means that the proposed security mechanism efficiently withstands a side-channel attack.

**5.5.2 Resists insider attacks.** The registration center (RC) first picks a random number s of size 160-bits, $r_{TS}$, $x_{TS}$, then concatenates it with the identity of a mobile user or teleworking server to quickly calculate the secret session key. The hash code generated is a collision-free and non-readable format, so the $\mathcal{A}$ cannot, at any stage, identify the identity or password from stored values. Similarly, the messages exchanged among participants are changed for upcoming sessions; if an $\mathcal{A}$ gets access to the internal credentials, their attempt fails due to choosing large random numbers by each peer, complex calculations, and arbitrary computation of the secret session key. Therefore, the teleworking environment security mechanism resists insider threats.

**5.5.3 Resists stolen-verifier attack.** Suppose an attacker $\mathcal{A}$ steals the mobile device, embezzles personal values from the teleworking server's memory and attempts to figure out the identities or passwords. In that case, they will fail because the proposed security mechanism is based purely on large random numbers and SHA-1. Due to the large random numbers $r_1$, $r_2$, $r_3$, $r_4$, $r_5$, s, $x_{TS}$, $r_{TS}$ for every session key and the linkage of these numbers with the identity/password, $\mathcal{A}$ cannot succeed. Therefore, the proposed scheme withstands a stolen verifier attack.

**5.5.4 Resists man-in-the-middle attack.** Suppose an $\mathcal{A}$ attempts to modify, discard, update, copy, or divert the exchanged information between MU→RC ($B^/$, $r_1^/$, F, G, $T_1$), RC→TS ($r_1^/$, $A^{//}$, $Q_2$, $Q_3$, $T_2$), or TS→RC ($r_3^/$, $Q_4$, $Q_5$, $T_3$), RC→MU ($r_3^/$, $Q_5$, $Q_6$, $Q_7$, $T_4$) and the peers believed that some malicious entity acted, then promptly stop the establishment of secure session key. But to do so, the $\mathcal{A}$ doesn't know the 160-bit long random numbers $r_1$, $r_2$,

$r_3$, $X_{MU}$, $r_{MU}$, $x_{TS}$, $r_{TS}$, so any illegal attempt will promptly be detected due to randomness in the exchanged information. Also, the recorded time threshold for each round trip and message confirmation can ensure mitigation of man-in-the-middle attack. Any independent connection establishment by any third party, the secret values which are not known to anyone, identities and random numbers can mitigate/cater to malicious deeds (man-in-the-middle attack, etc.) in the proposed lightweight authentication and key establishment protocol. Therefore, our proposed scheme is safe against man-in-the-middle attacks.

**5.5.5 Spoofing attack.** Suppose an $\mathcal{A}$ desires to succeed in sending false messages to a teleworking server (TS). In that case, they will fail due to verifying a pre-defined time threshold in each round trip. Also, other checks for different messages like $G = h(ID_{TS}||ID_{MU}||r_1||T_1)$ and $G' = h(ID_{TS}||ID_{MU}||r_1||T_1)$, $Q_3 = h(h(A||r_2||ID_{TS}||T_2)$ and $Q_3' = h(h(A||r_2||ID_{TS}||T_2)$ in the proposed security mechanism can strongly mitigate a spoofing attack.

**5.5.6 Withstands replay attack.** If an $\mathcal{A}$ capture $(B', r_1', F, G, T_1)$ message or $(r_1', A'', Q_2, Q_3, T_2)$ or $(r_3', Q_4, Q_5, T_3)$ or $(r_3', Q_5, Q_6, Q_7, T_4)$ or all form the open network channel and reply some other time for a potential replay attack. $\mathcal{A}$ don't make successful due to timestamp check in the first stage and several further checks, i.e., $G = h(ID_{TS}||ID_{MU}||r_1||T_1)$ and $G' = h(ID_{TS}||ID_{MU}||r_1||T_1)$, $Q_3 = h(h(A||r_2||ID_{TS}||T_2)$ and $Q_3' = h(h(A||r_2||ID_{TS}||T_2)$ in the second stage. Therefore, the proposed security mechanism is robust against reply attacks.

**5.5.7 Key secrecy.** Each entity mutually authenticate each other like $G = h(ID_{TS}||ID_{MU}||r_1||T_1)$ and $G' = h(ID_{TS}||ID_{MU}||r_1||T_1)$, $Q_3 = h(h(A||r_2||ID_{TS}||T_2)$ and $Q_3' = h(h(A||r_2||ID_{TS}||T_2)$ which in turn confirming secret session key $SK = h(h(A||r_2||r_1||r_3||A')$, $Q'_5 = h(SK||h(A||r_2)||r_1)$. This means that the secrecy of the secret session key is strong in the proposed security mechanism.

**5.5.8 Password guessing attack.** Suppose an adversary $\mathcal{A}$ desires to extract the internal secret parameters from the mobile devices due to $D = (A'||B')\oplus h(ID_{MU}||PW||r_1)$, $E = ID_{MU}\oplus h(D||PW||r_1)$, $r_1' = r_1\oplus h(ID_{TS}||ID_{MU})$, $F = ID_{TS}\oplus h(ID_{MU}||T_1)$, $G = h(ID_{TS}||ID_{MU}||r_1||T_1)$ and 64-bits of random numbers $r_{MU}$, $x_{MU}$, $r_1$, he/she cannot extract password from the stolen mobile device or illegal internal entry. Also, the proposed protocol has many checks at different round trips and can also deny any illegal attempt of an $\mathcal{A}$ for guessing online/offline password guessing attacks. Therefore, the proposed protocol resists password-guessing attacks.

**5.5.9 Untracebality.** The user in the proposed protocol is untraceable because the messages transmitted over a public network channel cannot be revealed to an attacker. For example, if an attacker desires to find the identity from these messages $\{B', r_1', F, G, T_1\}$, $\{r_1', A'', Q_2, Q_3, T_2\}$, $\{r_3', Q_4, Q_5, T_3\}$, $\{r_3', Q'_5, Q_6, Q_7, T_4\}$ they have to pass from many complex calculations. Also, the timestamp, random nonce and 64 bits of long key are concatenated with the identity, and the attacker couldn't figure out any useful information from any of the messages exchanged over public channels. Therefore, the proposed protocol provides the facility for the user to be untraced and secure, and their privacy could be preserved.

**5.5.10 Physical protection.** If someone wants to avail of the facilities using the proposed protocol must pass the registration phase. Suppose some culprit captures any legitimate user or mobile device and tries to find valuable credentials from it. The culprit must compute $A = h(ID_{TS}||r_{TS})$, $B = h(ID_{TS}||s)$, $C = h(IDTS||x_{TS})$ and $D = r_{TS}\oplus C$ for the mobile device, and $A' = h(ID_{MU}||r_{MU})$, and $B' = A/\oplus h(s||x_{MU})$ for the teleworking server; which is absolutely not possible. Suppose someone finds useful credentials, like key, identity, or any other parameter, they cannot establish a connection with the system due to the dynamicity of each message, and such an attempt could promptly be highlighted to the organization, and no peer can negotiate with an illegitimate third party at any stage. Therefore, the physical protection feature is available in our scheme.

**5.5.11 Withstands DoS attack.** If an $\mathcal{A}$ capture (B$'$, $r_1'$, F, G, T$_1$) message or ($r_1'$, A$''$, Q$_2$, Q$_3$, T$_2$) or ($r_3'$, Q$_4$, Q$_5$, T$_3$) or ($r_3'$, Q$_5$, Q$_6$, Q$_7$, T$_4$) or all form the open network channel and launches a DoS attack on the system. $\mathcal{A}$ don't make successful due to timestamp check in the first stage and several further checks, i.e., G = h(ID$_{TS}$||ID$_{MU}$||$r_1$||T$_1$) and G$'$ = h(ID$_{TS}$||ID$_{MU}$||$r_1$||T$_1$), Q$_3$ = h(h(A||$r_2$||ID$_{TS}$||T$_2$) and Q$_3'$ = h(h(A||$r_2$||ID$_{TS}$||T$_2$) in the second stage. Therefore, the proposed security mechanism is robust against DoS attacks.

**5.5.12 Perfect forward secrecy.** We have a user password change phase, in which a legal user can easily, securely and efficiently update their password without interacting with the TS or RC, which means the proposed secure framework offers high scalability and deposits the RC. It means any change in the stored credentials could be submitted to RC and alternately to all peers. So, upon computing the session key, all the credentials can be secretly changed from the user side to the server. This prominent feature is available in the proposed security framework.

# 6 Performance and comparative analyses

The performance of the proposed protocol can be measured by considering computation and communication costs. These are as follows:

## 6.1 Computation costs analysis

Suppose T$_E$ is the time consumed when a random number is extracted, T$_h$ is the time for a collision-free one-way hash function, and T$_{XoR}$ is the time of XOR operation, as shown in Table 4 and diagrammatically is shown in Fig 7. According to [38,39], the different times for different cryptographic operations are as follows:

- Computation time for extracting random numbers T$_E$ = 2.011ms

- Computation time for one-way hash function T$_h$ = 0.09ms

- Computation time for bit-wise XOR Operation T$_{XoR}$≈0ms

It is worth mentioning that only the cost calculated in the login authentication phase can be considered as computation costs, which is 6T$_E$+42T$_h$+18T$_{XOR}$ = 6(2.011)+42(0.09)+0 = 12.066 +3.78 = 15.786ms, while the computation costs of registration phase should be discarded.

**Table 4. Computation cost in milliseconds.**

| Phase | Participants | Operations | Total Operations | Costs | Total Costs |
|---|---|---|---|---|---|
| Registration | MU→RC | 2T$_E$+3T$_h$+1T$_{XOR}$ | 4T$_E$+5T$_h$+2T$_{XOR}$ | 4(2.011)+5(0.09)+0 = 8.044+0.45 = 8.494ms | 15.78 ms |
| | TS→RC | 2T$_E$+2T$_h$+1T$_{XOR}$ | | | |
| Authentication | MU | 2T$_E$+14T$_h$+6T$_{XOR}$ | 2(2.011)+14(0.09)+0 = 4.154+1.26 = 5.414ms | 6T$_E$+42T$_h$+18T$_{XOR}$ = 6(2.011)+42(0.09)+0 = 12.066+3.78 = 15.786ms | |
| | RC | 2T$_E$+16T$_h$+8T$_{XOR}$ | 2(2.011)+16(0.09)+0 = 4.154+1.44+0 = 5.594ms | | |
| | TS | 2T$_E$+12T$_h$+4T$_{XOR}$ | 2(2.011)+12(0.09)+0 = 4.154+1.08+0 = 5.234ms | | |

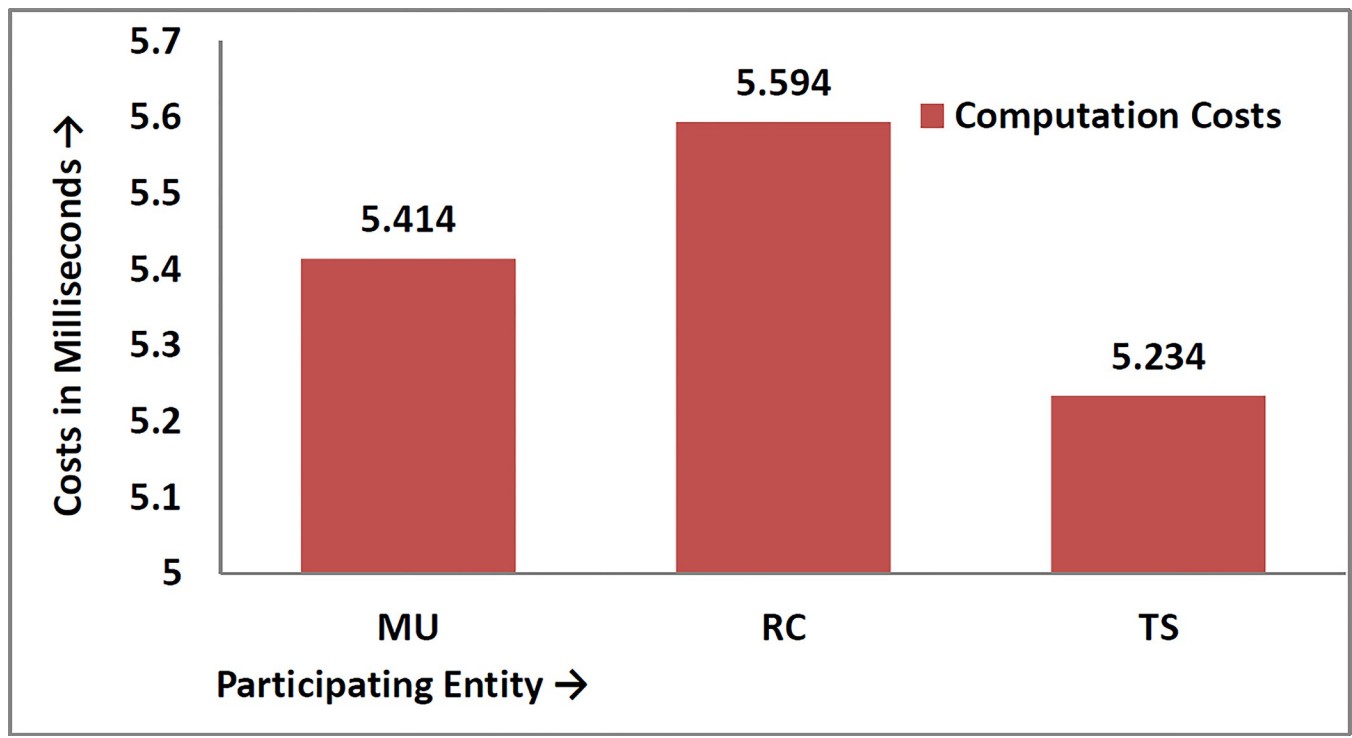

**Fig 7. Computation costs in milliseconds.**

## 6.2 Communication costs analysis

The messages transmitted during the login and authentication phase among different participating entities are $\{B', r_1', F, G, T_1\}$, $\{r_1', A'', Q_2, Q_3, T_2\}$, $\{r_3', Q_4, Q_5, T_3\}$, and $\{r_3', Q_5, Q_6, Q_7, T_4\}$. According to [38,39], identity takes 60 bits of space, timestamp 56, random number 64 bits, and one-way to function is 156 bits of memory space; then the communication costs of the proposed protocol in bits are shown in Table 5, while diagrammatically as shown in Fig 8.

## 6.3 Comparative analysis

By comparing the proposed protocol in terms of extra security features with Chall et al. [22], Wazid et al. [23], Shuai et al. [24], and Oh et al. [25], which is shown in Table 6. The result shows that our protocol is resisting all the design goals discussed in section II of the paper.

Similarly, to compare the proposed scheme with Xia et al. [7], Wazid et al. [23], Ding et al. [26], and Yang et al. [40] in terms of computation and communication costs, as shown in Table 7. The results show that the proposed security mechanism is lightweight, fast, secure, and suitable for practical implementation in the teleworking environment. The comparison in

**Table 5. Communication costs in bits.**

| Participants | Message Exchanged | Costs | Costs/Participants | Total Costs |
|---|---|---|---|---|
| MU→RC | $\{B', r_1', F, G, T_1\}$ | 156+156+156+156+56 | 680 | **2564 Bits** |
| RC→TS | $\{r_1', A'', Q_2, Q_3, T_2\}$ | 156+156+156+156+56 | 680 | |
| TS→RC | $\{r_3', Q_4, Q_5, T_3\}$ | 156+156+156+56 | 524 | |
| RC→MU | $\{r_3', Q_5, Q_6, Q_7, T_4\}$ | 156+156+156+156+56 | 680 | |

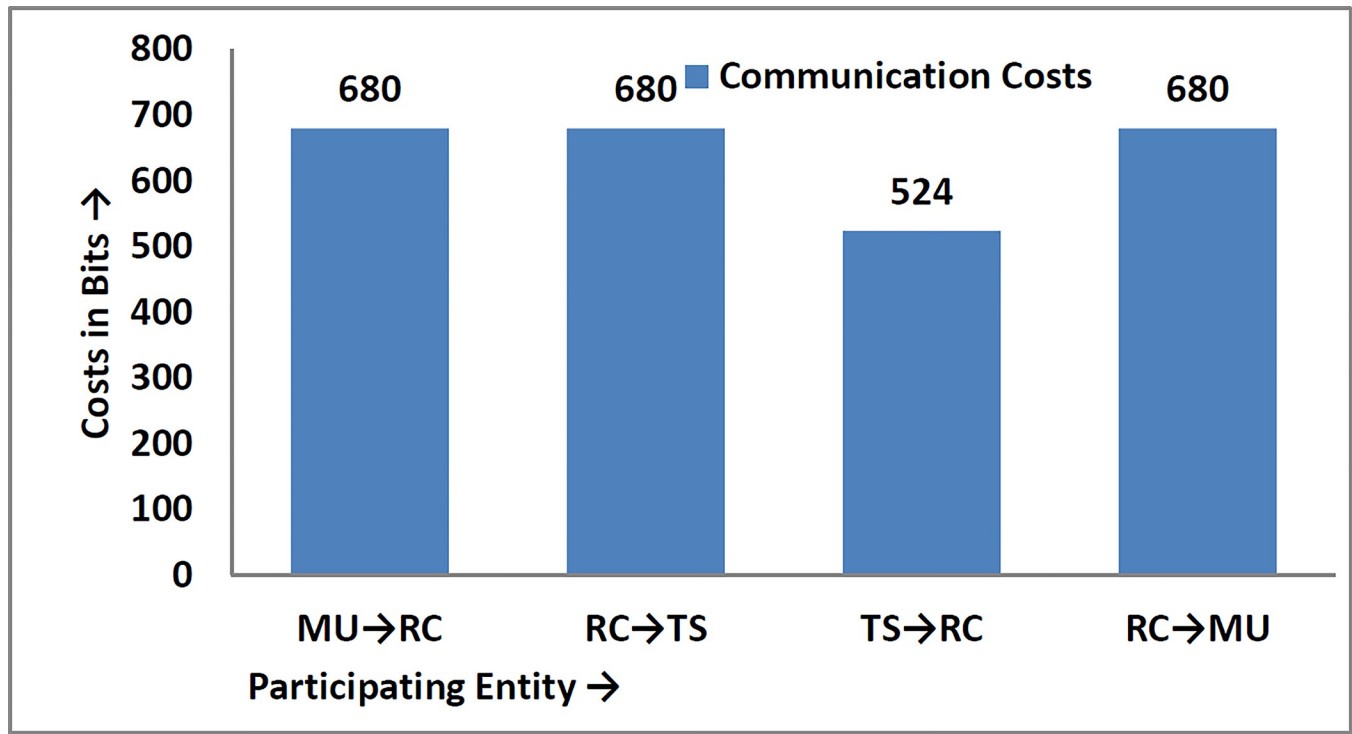

**Fig 8. Communication costs in bits.**

**Table 6. Security functionalities comparison.**

| Goals↓ / Schemes→ | [22] | [23] | [24] | [25] | Our |
|---|---|---|---|---|---|
| G1: Message authentication and integrity | Y | Y | Y | Y | Y |
| G2: Confidentiality and Authorization | Y | Y | Y | Y | Y |
| G3: Conditional Privacy-Preserving | Y | × | Y | × | Y |
| G4: Untraceability | × | Y | Y | Y | Y |
| G5: Physical Protection | Y | Y | × | Y | Y |
| G6: Resilience to Insider Threat | Y | × | Y | Y | Y |
| G7: Mutual Authentication | × | × | × | Y | Y |
| G8: Perfect Forward Secrecy | Y | Y | × | Y | Y |
| G9: Resists Man-in-the-Middle Attack | Y | Y | Y | × | Y |
| G10: Resists Denial-of-Service (DoS) Attack | Y | Y | Y | Y | Y |

Whereas Y means the scheme successfully achieves the given goal.

×The method doesn't achieve the mentioned security goal.

**Table 7. Performance metrics comparison.**

| Schemes→ / Performance Metrics↓ | [7] | [23] | [26] | [40] | Proposed |
|---|---|---|---|---|---|
| **Communication Costs in Bits** | 2816 | 3232 | 4096 | 3840 | 2564 |
| **Computation Costs in Milliseconds** | 58.828 | 46.54 | 16.42 | 47.095 | 15.78 |

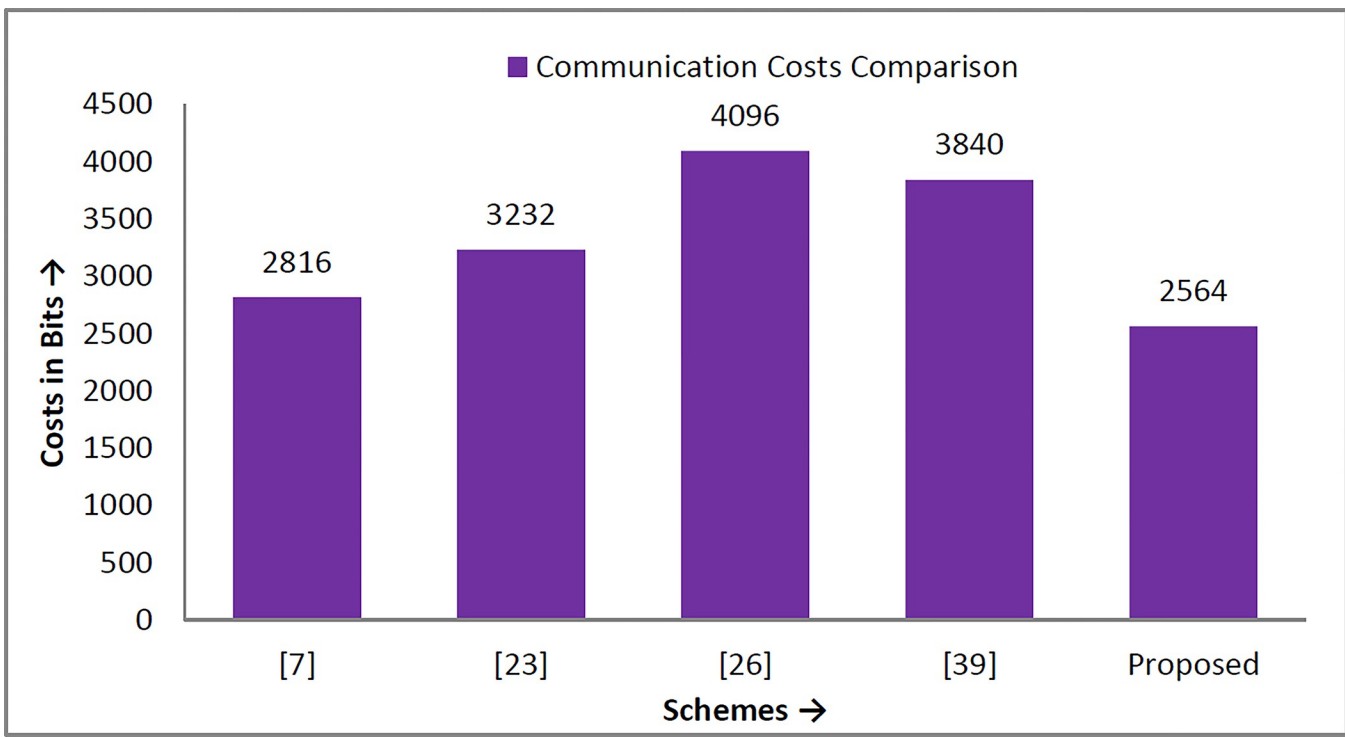

**Fig 9. Communication cost comparison graph.**

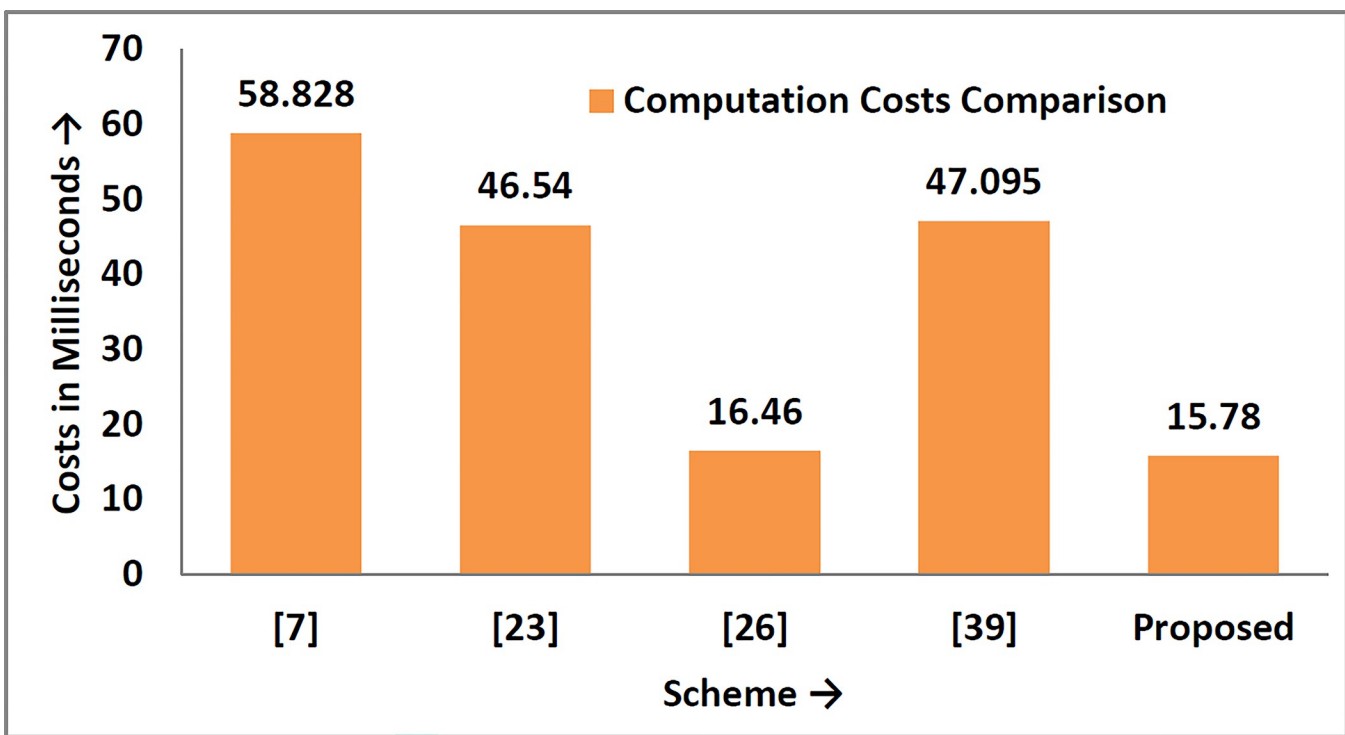

**Fig 10. Computation cost comparison graph.**

**Table 8. Percentage improvement in our protocol.**

| Schemes→ | [7] | [23] | [26] | [40] |
|---|---|---|---|---|
| **% Improvement in Communication Costs** | $= \frac{2816-2564}{2816} x\ 100$ <br> $= 8.94\%$ | $= \frac{3232-2564}{3232} x\ 100$ <br> $= 20.66\%$ | $= \frac{4096-2564}{4096} x\ 100$ <br> $= 34.4\%$ | $= \frac{3840-2564}{3840} x\ 100$ <br> $= 33.22\%$ |
| **% Improvement in Computation Costs** | $= \frac{58.828-15.78}{58.828} x\ 100$ <br> $= 73.17\%$ | $= \frac{46.54-15.78}{46.54} x\ 100$ <br> $= 66.09\%$ | $= \frac{16.42-15.78}{16.42} x\ 100$ <br> $= 3.89\%$ | $= \frac{47.095-15.78}{47.095} x\ 100$ <br> $= 66.49\%$ |

computation and communication overheads are diagrammatically depicted in Figs 9 and 10, respectively.

Furthermore, the proposed protocol has better performance than its competitors, i.e. Xia et al. [7], Wazid et al. [23], Ding et al. [26], and Yang et al. [40]. The percentage improvement of the proposed protocol with the mentioned schemes is shown in Table 8.

Table 8 argued that the proposed protocol is 8.94% better in communication costs than the Xia et al. [7] scheme, 20.66% from the Wazid et al. [23] scheme, 34.4% from the Ding et al. [26] and 33.22% from the Yang et al. [40] scheme. The maximum improvement in communication costs is 34.4%, and the minimum is 8.94%. Similarly, the percentage improvement of our scheme in terms of computation costs is 73.17% with Xia et al. [7], 66.09% with Wazid et al. [23], 3.89% with Ding et al. [26] and 66.49% with Yang et al. [40] scheme. The maximum improvement is 73.17%, and the minimum is 3.89%. It is keeping in view that our scheme is superior to its competitors.

## 7 Conclusion

Nowadays, people prefer remote work through the Internet (teleworking environment) instead of visiting physically. Such an environment is prone to numerous security issues like eavesdropping, unavailability, masquerading, replay, DoS, etc., and attacks. To make it prone-free, we have, in this article, proposed a lightweight and robust authentication protocol for those continuously using the Internet for remote monitoring of official work and securely authenticating before starting work. We have used SHA1 for the design of a protocol that is lightweight, robust in security, and offers efficient services to all the participants. We have tested the security of the proposed authentication protocol through a well-known formal technique, ROM analysis, programming verification toolkit ProVerif2.03, and informally via programmatic discussions. The performance analysis of the proposed security mechanism has been evaluated by considering computation and communication costs. Upon comparing the proposed mechanism with the existing security schemes in terms of security features and performance metrics, it has been demonstrated that the proposed protocol achieved a maximum of 34% improvement in computation and 73% improvement in communication costs and resisted all known vulnerabilities and can be implemented practically in the teleworking environment.

In the future, the proposed security scheme can be designed using blockchain technology, and the simulation part can be conducted through a network security simulator (*NeSSi*).

## Supporting information

**S1 Appendix. A ProVerif2.03 code.**
(TXT)

## Acknowledgments

Thanks to the deanship of research, Univeristy of Bisha, for continuous support.

## Author Contributions

**Conceptualization:** Fahad Algarni.

**Data curation:** Saeed Ullah Jan.

**Formal analysis:** Saeed Ullah Jan.

**Funding acquisition:** Fahad Algarni.

**Investigation:** Saeed Ullah Jan.

**Methodology:** Saeed Ullah Jan.

**Project administration:** Fahad Algarni.

**Resources:** Fahad Algarni.

**Software:** Fahad Algarni.

**Supervision:** Fahad Algarni.

**Validation:** Saeed Ullah Jan.

**Writing – original draft:** Saeed Ullah Jan.

**Writing – review & editing:** Saeed Ullah Jan.

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
