## [Decision Letter · Decision Letter 0]

21 Dec 2023

PONE-D-23-37453A Lightweight and Secure Protocol for Teleworking EnvironmentPLOS ONE

Dear Dr. Algarni,

Thank you for submitting your manuscript to PLOS ONE. After careful consideration, we feel that it has merit but does not fully meet PLOS ONE’s publication criteria as it currently stands. Therefore, we invite you to submit a revised version of the manuscript that addresses the points raised during the review process.

Based on the comments of the reviewers, I recommend this paper for major revision. ==============================Comments from PLOS Editorial Office: We note that one or more reviewers has recommended that you cite specific previously published works. As always, we recommend that you please review and evaluate the requested works to determine whether they are relevant and should be cited. It is not a requirement to cite these works. We appreciate your attention to this request.

We look forward to receiving your revised manuscript.

Kind regards,

Pandi Vijayakumar, Ph.D

Academic Editor

PLOS ONE

Journal Requirements:

2. Please upload a copy of Supporting Information Figure/Table/etc. S2 which you refer to in your text on page 28.

Reviewers' comments:

Reviewer's Responses to Questions

**Comments to the Author**

1. Is the manuscript technically sound, and do the data support the conclusions?

Reviewer #1: Yes

Reviewer #2: Yes

Reviewer #3: No

2. Has the statistical analysis been performed appropriately and rigorously? 

Reviewer #1: Yes

Reviewer #2: Yes

Reviewer #3: No

3. Have the authors made all data underlying the findings in their manuscript fully available?

Reviewer #1: Yes

Reviewer #2: Yes

Reviewer #3: No

4. Is the manuscript presented in an intelligible fashion and written in standard English?

Reviewer #1: Yes

Reviewer #2: Yes

Reviewer #3: No

5. Review Comments to the Author

Reviewer #1: 1- Please mention numerical values that confirm the efficiency of the proposed system in the abstract

2- No keywords were mentioned

3- Organizing the relevant research listed in the introduction according to the year of publication from oldest to newest, as is the case for references in Table 1

4- Comparison was made with references published within the period 2017-2021. Why was the comparison not made with a reference published during the year 2023, if possible?

5- Inserting numerical values from the tables and figures obtained into the results to build conclusions based on more realistic values.

6- Unifying the format of references, in addition to including recent references published during the year 2023

Reviewer #2: The manuscript under the titled “A Lightweight and Secure Protocol for Teleworking Environment” proposed the work on lightweight and secure technique for teleworking. This study tried to resist against various attacks may be performed on the system, the comparative analysis is encouraging and sufficient.

All the review comments are uploaded.

Reviewer #3: The article proposes a robust security protocol for remote work environments, emphasizing teleworking security. The protocol undergoes a thorough security analysis, employing tools such as ROM analysis and ProVerif2.03 simulation. Extensive cryptographic functions, biometrics handling, and secure communication channels are key features. The article highlights the protocol's performance through computation and communication cost analyses, comparing favorably with existing schemes. Future considerations include potential implementation with blockchain technology and network security simulation.

Clarity and Structure:

• Introduction (Section Number Missing): The section number for the introduction is not specified. Clarify this for better reference.

• Introduction (General Paragraph): Consider revising the first paragraph of the introduction to align more closely with the research objectives.

• Introduction (Technology and Human Schedule): The sentence, "In an age where technology is getting smarter, the schedule of the human being is equally becoming dense," is unclear. Consider revising for better clarity.

• Introduction (Abbreviation IoT): The author has mentioned network-enabled devices (IoT) in the introduction section. Confirm if this is the correct abbreviation and if it's appropriately introduced.

Technical Content:

Proposed Protocol:

• Complexity and Length: The protocol is complex and lengthy. Consider simplifying without sacrificing security.

• Cryptography Functions: Ensure cryptographic functions are secure and comply with current standards.

Registration Phase:

• Use of XOR: Verify the secure use of XOR operations in the protocol.

• Biometrics Handling: Thoroughly review the handling of biometrics (Gen() and Rep() functions) to prevent false rejections or acceptances.

• Login and Authentication Phase:

• Time Validation: Ensure proper time validation considering clock synchronization issues.

• Random Number Generation: Verify the security of random number generation.

Cryptanalysis of Baseline Scheme:

• Vulnerabilities: Address vulnerabilities identified in the cryptanalysis.

Proposed Protocol (Continued):

• Notation Clarity: Ensure clear and consistent notation throughout the section.

• Modular Design: Consider breaking down the protocol into smaller, modular components for readability.

• Detailed Security Analysis: Conduct a thorough security analysis, possibly using formal methods or involving a third party.

• Standard Compliance: Ensure compliance with established security standards and best practices.

• Clarify Assumptions: Clearly state any assumptions about the system, environment, or adversary model.

• Considerations for Real-World Implementation: Address practical issues like key management, secure channel establishment, and real-world constraints.

Writing Style and Grammar:

• Consistency in Notation: Ensure consistent notation throughout the section. For example, you have both RC and RC�. Make sure to use a consistent notation for entities and variables.

• Punctuation and Sentence Structure: Simplify lengthy sentences for better clarity.

Security Analysis:

• Equation Clarification: Provide a more precise description for "attacker(SK[])" in line 462.

GNY Logic Analysis:

• Context for Formulas and Statements: Provide context for how formulas and statements in Table 3 are applied in subsequent sections.

Informal Security Analysis:

• Feedback on Assumptions: Explicitly state any specific conditions or constraints for assumptions.

• Consistency in Formatting: Ensure consistent formatting for equations and queries.

• Context for Equations and Statements: Provide a bit more context for each equation or statement for better understanding.

Performance and Comparison Analysis:

• Line number 605 (Computation Costs): Ensure consistency in Total Costs with individual costs in line number 605.

• Line number 608-611 (Communication Costs): Check the consistency of the values in bits and the total costs. The values seem to be additive, but it's important to ensure correctness.

• Table 6 (Security Functionalities Comparison): Briefly explain the significance of each security goal (G1 to G10) for better understanding.

• Table 7 (Performance Comparison): Double-check units for Communication Costs and clarify Computation Costs.

• Figures 3(a) and 3(b): Ensure figures are clear and correctly labeled, with separate files as indicated in the text.

• Present visual comparative graphs that illustrate the efficiency of the proposed system in relation to existing schemes.

Conclusion:

• Conclusion Recap: Consider adding a brief recap of the main findings or contributions.

• Future Work Direction: The mention of future work involving blockchain technology and a network security simulator is a positive addition. It provides a direction for future research.

Proofreading: Conduct a thorough proofread for grammatical errors and typos.

6. PLOS authors have the option to publish the peer review history of their article (what does this mean?). If published, this will include your full peer review and any attached files.

Reviewer #1: No

Reviewer #2: **Yes: **Jitendra Vikram Tembhurne

Reviewer #3: No

---

## [Author Response · Author response to Decision Letter 0]

2 Jan 2024

We have addressed each point of the reviewers... and rebuttal letter namely "Response to Reviewers Comments" is attached.

---

## [Decision Letter · Decision Letter 1]

9 Jan 2024

PONE-D-23-37453R1A Lightweight and Secure Protocol for Teleworking EnvironmentPLOS ONE

Dear Dr. Algarni,

Thank you for submitting your manuscript to PLOS ONE. After careful consideration, we feel that it has merit but does not fully meet PLOS ONE’s publication criteria as it currently stands. Therefore, we invite you to submit a revised version of the manuscript that addresses the points raised during the review process.

Based on the comments of the reviewers, I recommend this paper for major revision. 

We look forward to receiving your revised manuscript.

Kind regards,

Pandi Vijayakumar, Ph.D

Academic Editor

PLOS ONE

Journal Requirements:

Additional Editor Comments :

Based on the comments of the reviewers, I recommend this paper for minor revision.

Reviewers' comments:

Reviewer's Responses to Questions

**Comments to the Author**

1. If the authors have adequately addressed your comments raised in a previous round of review and you feel that this manuscript is now acceptable for publication, you may indicate that here to bypass the “Comments to the Author” section, enter your conflict of interest statement in the “Confidential to Editor” section, and submit your "Accept" recommendation.

Reviewer #1: All comments have been addressed

Reviewer #3: All comments have been addressed

2. Is the manuscript technically sound, and do the data support the conclusions?

Reviewer #1: Yes

Reviewer #3: Yes

3. Has the statistical analysis been performed appropriately and rigorously? 

Reviewer #1: Yes

Reviewer #3: Yes

4. Have the authors made all data underlying the findings in their manuscript fully available?

Reviewer #1: Yes

Reviewer #3: Yes

5. Is the manuscript presented in an intelligible fashion and written in standard English?

Reviewer #1: Yes

Reviewer #3: Yes

6. Review Comments to the Author

Reviewer #1: (No Response)

Reviewer #3: The corrections have been updated in the manuscript; however, some major corrections are still required in the article, as outlined below.

Notations: The article introduces numerous notations (e.g., LA1, G1, B/, r1/) without representation in the notations table (Table 1). It is crucial to include all notations in the table for clarity and reference throughout the article.

Performance Metrics Table: In Table 7 (Performance Metrics Comparison), the cost should be mentioned along with its units (e.g., bits). This addition will enhance the comprehensibility of the presented metrics.

Figures 1 and 2: Figures 1 and 2 appear blurred, necessitating a revision to improve visual clarity. Enhanced image quality will contribute to better understanding.

Figures 3 and 4: In Figures 3 and 4, consider either writing the x-axis in full form or placing the full form inside the graph to eliminate potential confusion.

Figure 2 - System Model: Provide component names in Figure 2's system model. Additionally, there is a discrepancy between mentioning "mobile user" and depicting a PC in the image. Clarification or correction is needed for consistency.

Figures 5 and 6: In Figures 5 and 6, it is essential to mention the units of the represented data. Clearly stating the units will enhance the interpretation of the figures.

Goals Achievement: The authors state that Goals 1, 2, and 3 are achieved. However, the article lacks information on the accomplishment of Goals 4 to 10. Providing insights into the fulfilment of all stated goals is necessary for a comprehensive understanding.

Reference: The title reference is more suitable to the proposed system "Dual Authentication and Key Management Techniques for Secure Data Transmission in Vehicular Ad Hoc Networks". Consider addition to the reference section.

7. PLOS authors have the option to publish the peer review history of their article (what does this mean?). If published, this will include your full peer review and any attached files.

Reviewer #1: No

Reviewer #3: No

---

## [Author Response · Author response to Decision Letter 1]

10 Jan 2024

A rebuttal letter is attached in which reviewer 1 fully accepted our previous efforts while reviewer 3 has put some useful additional concerns which we have addressed up to a maximum extent. Now, the paper looks both thematic and structural unity summing making the work creditable. Hope, you will accept our efforts.

Regards........!

Prof; Fahad

---

## [Decision Letter · Decision Letter 2]

23 Jan 2024

A Lightweight and Secure Protocol for Teleworking Environment

PONE-D-23-37453R2

Dear Dr. Algarni,

We’re pleased to inform you that your manuscript has been judged scientifically suitable for publication and will be formally accepted for publication once it meets all outstanding technical requirements.

Kind regards,

Pandi Vijayakumar, Ph.D

Academic Editor

PLOS ONE

Additional Editor Comments (optional):

Reviewers' comments:

Reviewer's Responses to Questions

**Comments to the Author**

1. If the authors have adequately addressed your comments raised in a previous round of review and you feel that this manuscript is now acceptable for publication, you may indicate that here to bypass the “Comments to the Author” section, enter your conflict of interest statement in the “Confidential to Editor” section, and submit your "Accept" recommendation.

Reviewer #1: All comments have been addressed

Reviewer #3: All comments have been addressed

2. Is the manuscript technically sound, and do the data support the conclusions?

Reviewer #1: Yes

Reviewer #3: Partly

3. Has the statistical analysis been performed appropriately and rigorously? 

Reviewer #1: Yes

Reviewer #3: Yes

4. Have the authors made all data underlying the findings in their manuscript fully available?

Reviewer #1: Yes

Reviewer #3: Yes

5. Is the manuscript presented in an intelligible fashion and written in standard English?

Reviewer #1: Yes

Reviewer #3: Yes

6. Review Comments to the Author

Reviewer #1: (No Response)

Reviewer #3: Based on the revisions made by the authors, I recommend that the manuscript be accepted for publication. The changes have significantly strengthened the paper, and it now aligns well with the standards

7. PLOS authors have the option to publish the peer review history of their article (what does this mean?). If published, this will include your full peer review and any attached files.

Reviewer #1: No

Reviewer #3: **Yes: **RAJKUMAR S C

---

## [Editor Report · Acceptance letter]

11 Mar 2024

PONE-D-23-37453R2 

PLOS ONE

Dear Dr. Algarni, 

I'm pleased to inform you that your manuscript has been deemed suitable for publication in PLOS ONE. Congratulations! Your manuscript is now being handed over to our production team.

Kind regards, 

on behalf of

Dr. Pandi Vijayakumar 

Academic Editor

PLOS ONE